# Intelligent Go-Explore: Standing on the Shoulders of Giant Foundation Models

**Cong Lu**[1,2]
conglu@cs.ubc.ca

**Shengran Hu**[1,2]
srhu@cs.ubc.ca

**Jeff Clune**[1,2,3]
jclune@gmail.com

[1]University of British Columbia
[2]Vector Institute
[3]Canada CIFAR AI Chair

### Abstract

Go-Explore is a powerful family of algorithms designed to solve hard-exploration problems built on the principle of archiving discovered states, and iteratively returning to and exploring from the most promising states. This approach has led to superhuman performance across a wide variety of challenging problems including Atari games and robotic control, but requires manually designing heuristics to guide exploration (i.e., determine which states to save and explore from, and what actions to consider next), which is time-consuming and infeasible in general. To resolve this, we propose Intelligent Go-Explore (IGE) which greatly extends the scope of the original Go-Explore by replacing these hand-crafted heuristics with the intelligence and internalized human notions of interestingness captured by giant pretrained foundation models (FMs). This provides IGE with a human-like ability to instinctively identify how interesting or promising any new state is (e.g., discovering new objects, locations, or behaviors), even in complex environments where heuristics are hard to define. Moreover, IGE offers the exciting opportunity to *recognize and capitalize on serendipitous discoveries*—states encountered during exploration that are valuable in terms of exploration, yet where what makes them interesting was not anticipated by the human user. We evaluate our algorithm on a diverse range of language and vision-based tasks that require search and exploration. Across these tasks, IGE strongly exceeds classic reinforcement learning and graph search baselines, and also succeeds where prior state-of-the-art FM agents like Reflexion completely fail. Overall, Intelligent Go-Explore combines the tremendous strengths of FMs and the powerful Go-Explore algorithm, opening up a new frontier of research into creating more generally capable agents with impressive exploration capabilities. All our code is open-sourced at: `https://github.com/conglu1997/intelligent-go-explore`.

## 1 Introduction

Foundation models (FMs, Bommasani et al. (2021); OpenAI (2024); Brown et al. (2020); Team (2024); Touvron et al. (2023)) trained on giant internet-scale datasets have demonstrated strong general capabilities in reasoning (Wei et al., 2022) and understanding (Chang et al., 2024). As such, these models have been increasingly employed as autonomous agents (Liu et al., 2023; Yao et al., 2023b; Wang et al., 2024; Yao et al., 2023a; Shinn et al., 2023; Besta et al., 2024b) in decision-making tasks, showcasing the ability to adapt in-context (Dong et al., 2022; Olsson et al., 2022) to unseen tasks. Foundation models have also begun to see success in challenging reinforcement learning environments like Doom (de Wynter, 2024), real-world robotic control (Brohan et al., 2022; 2023) and 3D video games (Raad et al., 2024; Wang et al., 2023). However, a significant challenge remains: foundation model agents often struggle in environments that require deep exploration over extended time horizons (Liu et al., 2023). Overcoming this limitation would enable us to realize their potential as autonomous assistants in more open-ended domains like scientific discovery and innovation (Jiang et al., 2023). This paper introduces Intelligent Go-Explore (IGE), a novel

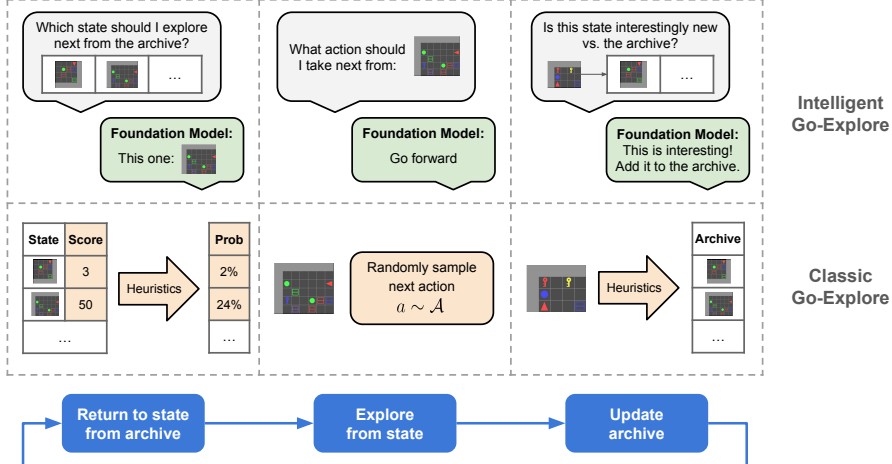

Figure 1: INTELLIGENT GO-EXPLORE (IGE) integrates the intelligence and internalized human notions of interestingness from giant pretrained FMs into all stages of the Go-Explore (Ecoffet et al., 2021a;b) algorithm, enabling FM agents to robustly explore in complex environments. **Bottom:** Classic Go-Explore solved hard exploration problems by archiving novel discovered states, resetting to promising ones via domain-specific heuristics, and then performing random exploration. **Top:** Our approach, INTELLIGENT GO-EXPLORE, enables Go-Explore to tackle virtually any type of problem that is representable in the context of a large language or multimodal model. Instead of manually defining heuristics, we query the foundation model at all stages, enabling our approach to automatically catch and return to serendipitous discoveries, and harness the power of FM agents to explore. The environment shown is the BabyAI game used in Section 4.2.

approach that combines the intelligence of foundation models with the powerful Go-Explore (Ecoffet et al., 2021a;b) framework to substantially increase the exploration capabilities of FM and reinforcement learning (RL, Sutton & Barto (2018)) agents.

Go-Explore is a popular family of algorithms in deep RL based on maintaining an archive of "interestingly new" discovered states and then iteratively returning to and exploring from the most promising states (see Figure 1 for an overview of the three stages). This framework has led to superhuman performance in a range of hard-exploration problems, including long-horizon Atari games and robotic control. However, the algorithm's success *largely relies on carefully hand-designed heuristics* at all three stages to guide exploration (Lu et al., 2024; Madotto et al., 2020). For example, in Montezuma's Revenge (Bellemare et al., 2013), an Atari game that was the previous grand challenge of exploration in deep RL, **(1)** saved states in the archive were returned to with probability proportional to factors like the number of times a state has been sampled before, **(2)** exploration was purely via random action sampling, and **(3)** the criteria for which states were considered interestingly new enough to be added to the archive depended on domain-specific factors like whether the agent visited a new location, or did so with more keys. Without this pre-specified knowledge, the quality of discovered trajectories is typically significantly worse (Ecoffet et al., 2021b).

These rigid, domain-specific choices are in stark contrast to human-like exploration of a new game, where players can often intuitively judge the value or interestingness of any particular state (Cooper, 2014). More importantly, *it is often impossible to know what is interesting or possible ahead of time* in complex domains. In the words of Isaac Asimov—*The most exciting phrase to hear in science, the one that heralds new discoveries, is not "Eureka!" but "That's funny.".* With this motivation, IGE stands on the shoulders of giant foundation models and uses their intelligence to **(1)** act as a judge to identify the most promising states to return to and explore from, **(2)** select the best actions to take from a selected state, and **(3)** recognize and capitalize on *serendipitous discoveries*—states encountered during exploration that are valuable in terms of exploration, yet where what makes them interesting was not anticipated by the human user. By leveraging the foundation model's internalized notions of interestingness (Zhang et al., 2024), IGE can decide whether a new state is interestingly new enough to be added to the archive as a stepping stone for future exploration (Figure 1, top).

We demonstrate IGE's ability to reliably improve the exploration capabilities of FM agents on a diverse range of language and vision-based tasks that require search and exploration. These settings include tasks that require *commonsense reasoning, long-term planning and memory, and handling*

*partial observability*. IGE integrates well with various agent strategies, including few-shot and chain-of-thought-based prompting, demonstrates consistent improvements over baselines across diverse foundation models, and will likely only get better as the capabilities of FMs continue to improve. While IGE performs strongly all-around, some highlights from our evaluation include: IGE reaches 100% success rate on Game of 24 (Yao et al., 2023a), a standard mathematical reasoning and search problem, 70.8% faster than classic graph search. On the BabyAI domain, IGE enables standard foundation models to succeed with visual observations zero-shot. Moreover, on the TextWorld (Côté et al., 2018) Coin Collector domain, IGE is the only algorithm that succeeds in discovering long-horizon optimal solution paths, where prior state-of-the-art FM agent frameworks like Reflexion (Shinn et al., 2023) fail.

INTELLIGENT GO-EXPLORE simultaneously empowers foundation model agents to *reliably explore*, and reimagines the scope of Go-Explore to tackle virtually any type of problem, without being limited to hand-designed heuristics. These abilities will substantially improve our ability to develop more generally capable agents, and increase the range of tasks they can learn how to solve.

## 2 BACKGROUND

**Go-Explore for Hard-Exploration Problems.** Go-Explore (Ecoffet et al., 2021b;a) is a family of algorithms designed to solve hard-exploration (Ladosz et al., 2022) problems based on the principle of remembering and returning reliably to promising states. The classic setting builds an "archive" of novel states it discovers in an environment, where similar states are grouped in a single "cell". These cells are defined by heuristics like having the same visual observation when downsampled to low resolution. In the beginning, the archive only contains the initial state. We describe the overall structure of the algorithm in the same order as Figure 1 (bottom): At each iteration, **(1)** promising states are selected from the archive through domain-specific heuristics, e.g., probabilistically sampling states proportional to their progress through the environment or potential to lead to new states. The agent returns to that state by resetting using the simulator or via a goal-conditioned policy, and **(2)** a sequence of random actions is taken to explore from that state. **(3)** All discovered states deemed interestingly new by the cell representation heuristics are added to the archive, and the process repeats. The strength of Go-Explore is due to addressing two critical impediments to exploration: forgetting how to reach previously visited states (detachment) and failing to first return to a state before exploring from it (derailment) (Ecoffet et al., 2021a).

This approach leads to a collection of high-return trajectories being discovered, which may then be fed into an imitation learning (Hussein et al., 2017) algorithm to produce a policy that generalizes and is robust to stochasticity. We adopt similar assumptions as the original setting, by assuming an agent can return to a previously discovered state by restoring in the simulator (e.g., a reset function in an RL environment). This assumption may readily be relaxed by training a policy to return to a given state, or in the foundation model case, by simply prompting the model with a past trajectory.

**Large Language and Multimodal Foundation Models.** The combination of model scaling and training over internet-scale data has resulted in a wide variety of foundation models (Bommasani et al., 2021) that exhibit generalist capabilities. In this paper, we consider autoregressive large language models (LLMs, Brown et al. (2020); OpenAI (2024); Touvron et al. (2023)) which learn to generate text completions by modeling the conditional probability of a new token given the preceding tokens, $p(x_t|x_{<t}; \theta)$. This framework enables LLMs to not only generate coherent text but crucially also exhibit human-like abilities, including on commonsense knowledge questions (Talmor et al., 2019) and complex reasoning tasks (Wei et al., 2022). These models may also be extended to other input modalities such as images by tokenizing these inputs into the same space as the text (Zhu et al., 2023a). When prompting an FM with an instruction, the user may decide to do so with no related examples (zero-shot), with a few successful examples in related problems (few-shot, Brown et al. (2020)), or ask for a chain of reasoning (chain-of-thought, Wei et al. (2022)) before responding.

## 3 DRIVING EXPLORATION WITH GIANT FOUNDATION MODELS

In this section, we propose INTELLIGENT GO-EXPLORE (IGE) which reimagines the classic Go-Explore algorithm as described in Section 2 with the intelligence of giant pretrained foundation models. Specifically, we introduce FM intelligence for selecting which archived state to return to and

explore from, which action to take from each state, and deciding whether a state is interestingly new and should be archived. IGE's use of foundation models is closely related to FM-as-a-judge (Zheng et al., 2023b), but instead uses foundation models as proxies of human judgment of exploration choices in an environment rather than the output of generative models. We illustrate our resultant algorithm at the top of Figure 1 and provide full pseudocode in Algorithm 1.

Wherever we query the foundation model, we introduce the overall strategy of Go-Explore alongside a brief description of the current environment in the "system message" (high-level directive) displayed below. The brief descriptions for each environment we evaluate on in Section 4 are listed in Appendix B. In the following sections, we detail our prompting techniques at each stage of IGE. The previous prompt history is visible to the agent, which enables each component of IGE to communicate with each other. We provide precise details on how we parse responses in Appendix D.1.

---

**System Prompt.**

[Brief Description Of Environment]
You will be prompted to perform systematic exploration in the style of Go-Explore. An archive will be maintained of interesting states found. You will be prompted to:
- Select a state from the archive that is the most promising, i.e., likely to lead to a solution or more novel states.
- Explore from states intelligently, by picking new actions.
- For each new state, determine if the state is interestingly new and should be added to the archive.

---

## 3.1 SELECT STATE FROM ARCHIVE

The power to easily store and return to promising discovered states is crucial to Go-Explore's ability to reliably solve long-horizon exploration problems. IGE leverages the foundation model's internalized notions of interestingness (Zhang et al., 2024) to select the most promising state to return to from the archive (Figure 1, left). This is far more flexible than classic Go-Explore, which relied on hardcoded hand-crafted heuristics to determine cell sampling probabilities. An example prompt is shown below.

Examples of the discovered states are given in Table 1. We assign indices to these states in a list and ask the FM to select a numerical index. We define a budget of $N_{state}$ "state-expansions". Each state expansion is followed by a sequence of exploratory actions, which we describe in the next section.

---

**State Selection Prompt.**

Current state archive:
[Discovered states]

Select the most promising state.

---

## 3.2 EXPLORE FROM STATE

In order to effectively explore from a state selected in the previous section, we leverage the power of foundation model agents (Liu et al., 2023; Huang et al., 2022) to choose how to act in an environment. This vastly improves on the original Go-Explore's use of random action sampling. One of the key strengths of IGE is that it is **complementary** to various FM agent reasoning strategies, *including zero-shot, few-shot, or even chain-of-thought-based prompting* (Yao et al., 2023b). We demonstrate this flexibility in Section 4.

One point of departure from the classic Go-Explore is that we additionally maintain a state-conditional action history for each archived state, so that IGE can avoid repeating previously tested options. While this information may already be available in the entire history, this helps *avoid any recency bias that can occur with longer contexts* (Zhao et al., 2021). The action history can be easily reiterated in the prompt, or the prompt could display the remaining untested actions. We define a budget of exploratory actions per state expansion $N_{action}$, which is typically far shorter than the full horizon of the environment and represents a small number of trial actions. An example prompt is shown here.

---

**Action Selection Prompt.**

[Agent-Specific Prompt]
Current state:
[Current State]
Previously tried actions:
[Previous Action History]

Output the next action.

---

Table 1: We show that INTELLIGENT GO-EXPLORE can efficiently explore over a diverse set of environments with increasing difficulty. We showcase IGE on environments with both language and vision-based observations. For each environment, we provide an example observation, samples from the action space, and the horizon of the task in the environment.

| | Game of 24 | BabyAI (Text and Visual) | | TextWorld |
|---|---|---|---|---|
| **Problem Type** | mathematical reasoning and search | partially observable gridworld with language instructions | | partial observability, long-term planning and memory, and common sense |
| **Observation** | "Current state: (2 8 8 14)" | "Goal: unlock the red door. You see a wall 4 steps forward, You see a yellow box 2 steps left." (Text-based) | (Vision-based) | "You arrive in a pantry... You see a shelf. The shelf is wooden. On the shelf you can see flour..." |
| **Next Actions** | - 2 + 8 = 10 Next: (8 10 14) 
 - 8 / 2 = 4 Next: (4 8 14) 
 - 14 + 2 = 16 Next: (8 8 16) 
 ... | - turn left 
 - turn right 
 - go forward | - pick up 
 - open door 
 - drop | - go east 
 - cook potato with oven 
 - unlock door with key 
 ... |
| **Task Horizon** | 3 | 64 or 128 | | 25, 40 or 80 |

## 3.3 UPDATE ARCHIVE

IGE stores discovered states in an archive, allowing the algorithm to return and explore from those points. To encourage better exploration, the stored states should be interesting—either promising states relevant to the task that could lead to further stepping stones or novel states that differ significantly from existing ones. As in other stages of IGE, we leverage the foundation model's internalized notions of interestingness (Zhang et al., 2024) to evaluate the interestingness of each state.

While the original Go-Explore required extensive domain knowledge to determine interestingness, IGE avoids this requirement and manual labor, critically gaining the ability to recognize and capitalize on serendipitous discoveries that could not have been predicted ahead of time. In practice, we propose two options to filter discovered states after a sequence of exploratory actions. The first is to iterate through every new state and ask whether each one is interestingly new and should be added to the archive. The second is to first add all states and then ask the foundation model to remove the uninteresting states. We discuss this choice later in Section 4.3; the second form is preferable in larger environments where there is more need to explicitly deprecate earlier discoveries that have become irrelevant so as not to overload the archive. An example prompt for the first option is shown below.

By default, IGE implements the foundation model at all three stages of Go-Explore, but we rigorously analyze the relative importance of each component in Section 5. Our focus is on the discovery of solutions to hard-exploration problems. These solutions could potentially be used for downstream reinforcement learning or even improve the foundation model in subsequent tasks through in-context learning, representing exciting avenues for future research.

**Archive Filtering Prompt.**

Current state archive: [State Archive]
New state: [Current State]

Is this state interestingly new (a novel state that is relevant to the task or could lead to further stepping stones), such that it should be added to the archive?

## 4 EMPIRICAL EVALUATION

In this section, we evaluate INTELLIGENT GO-EXPLORE across a diverse set of text environments that require search and exploration. We demonstrate IGE's ability to handle partially observable and complex observation spaces (including both text and visual inputs), discover solutions involving long chains of actions, and effectively improve the ability of FM agents to explore. For all our experiments, we use GPT-4 (OpenAI, 2024), one of the current SOTA LLMs, as our foundation model. We compare IGE to random action sampling, a naïve LLM baseline, and two SOTA FM agents, ReAct (Yao et al., 2023b) and Reflexion (Shinn et al., 2023). All methods receive the same environment descriptions, observations, and use the same number of environment steps, ensuring a fair comparison. Naïve LLM simply queries the LLM for an action conditional on the interaction history. ReAct prompts the agent to output its reasoning before making a decision. Based on ReAct, Reflexion further conditions the agent on the previous attempted episode, asking the agent to learn

from its mistakes. We provide an overview of our environments in Table 1. Full hyperparameters are detailed in Appendix E.

## 4.1 GAME OF 24

We first demonstrate the effectiveness of IGE in a mathematical reasoning task, Game of 24 (Yao et al., 2023a). The goal is to perform basic arithmetic operations $(+, -, \times, /)$ starting from 4 numbers to obtain 24. For example, given input $(4, 9, 10, 13)$, a possible solution could be $(10 - 4) \times (13 - 9) = 24$. We formulate the problem as an MDP (Sutton & Barto, 2018), where actions represent a reduction of two numbers by an arithmetic operation—i.e., the above solution would be represented as the sequence of state transitions $(4, 9, 10, 13) \xrightarrow{10-4=6} (6, 9, 13) \xrightarrow{13-9=4} (6, 4) \xrightarrow{6\times4=24} (24)$. Therefore, IGE uses the FM to iteratively expand possible solution paths and archive promising ones to return to. The action space is the range of possible next operations, displayed in the same manner as in Yao et al. (2023a).

We evaluate IGE across 100 hard test problems in Figure 2, and additionally include the standard (unweighted) graph search algorithms depth-first search (DFS) and breadth-first search (BFS) as reference. Since the combinatorial complexity of the problem is at most $\binom{4}{2} \cdot \binom{3}{2} \cdot 4^3 = 1152$, graph search is guaranteed to find a solution within that many actions. The system prompts for both IGE and the LLM baselines contain *few-shot examples* with correct calculations on different starting numbers. IGE rapidly reaches 100% success rate, on average 70.8% faster than the next best baseline, depth-first search (DFS)— this improvement is statistically significant ($\chi^2$ test, $p < 0.05$) at 150 operations, where IGE has solved all problems. This success may be attributed to the fact that language models have internalized mathematical intuition and are likely to be able to identify promising pairs like $(6, 4)$ that could easily be multiplied together for a solution.

All LLM agent baselines (naïve LLM, ReAct, Reflexion) eventually plateau and even get beaten by the unintelligent DFS. This highlights the need for diverse action selection, which IGE enables. A final point of comparison we make is to Tree of Thoughts (ToT, Yao et al. (2023a)) which achieved 74% on Game of 24 within their evalu-

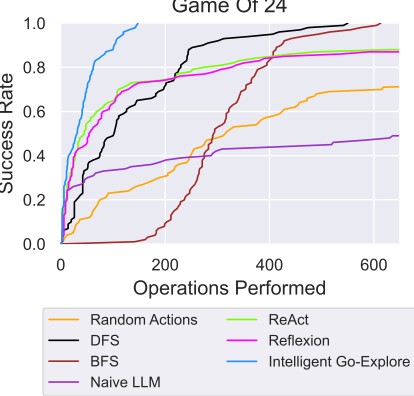

Figure 2: IGE explores the Game of 24 with the intelligence of FMs and reaches 100% success rate on average 70.8% faster than DFS, the next best baseline. IGE completes all problems within 150 environment operations. Our use of archiving and intelligent action selection allows us to greatly outperform prior LLM agents with an equal number of operations performed. The success rate is computed over 100 test problems.

ation budget. We emphasize that our evaluation setting is very different as IGE selects from the list of valid options rather than doing the math in context. However, we note the key difference to our method is that ToT evaluates and expands multiple reasoning paths following a tree structure, whereas IGE can easily jump around the search space—this is a crucial advantage in more complex environments (like those in the following sections), where it takes many coordinated actions to get from one state to another interesting state.

## 4.2 BABYAI TEXT AND VISUAL DOMAINS

Next, we show that IGE readily operates across multiple modalities in the BabyAI domains from Carta et al. (2023). The original domain is a procedurally-generated, partially-observable 2D grid-world with text-based observations. The agent is given a textual goal instruction which could correspond to one or more instructions in a sequence, e.g., "pick up X and then go to Y". As we can see from the observations in Table 1, the task is challenging even for humans to complete and requires forming a model of the world from partial observations. This kind of state observation would make it *hard to define heuristics to determine how good any particular state is*, as in classic Go-Explore. We additionally extend this to a **visual domain** by replacing text observations with partially observable image observations. IGE can naturally handle these by simply replacing text observations with images passed to the multimodal GPT-4o. The optimal path to a solution may include moving

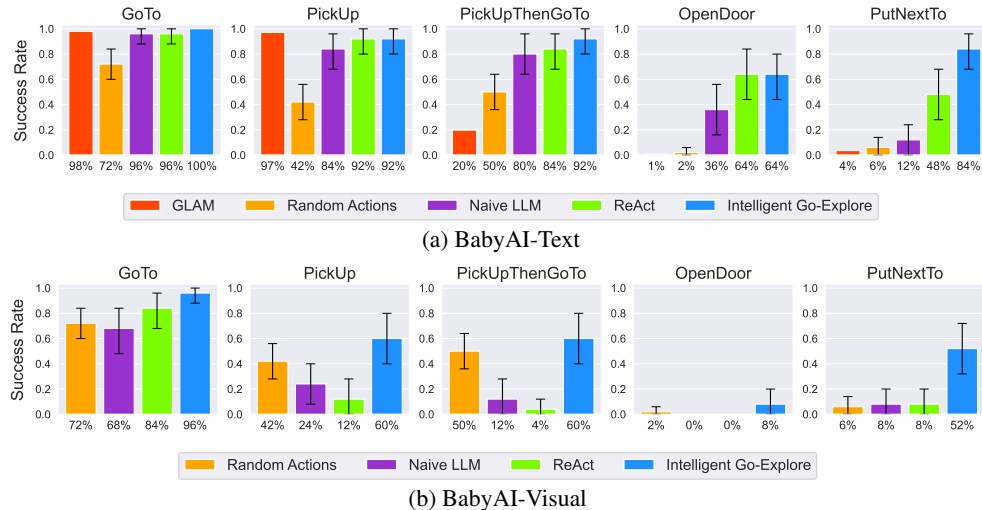

(a) BabyAI-Text

(b) BabyAI-Visual

Figure 3: IGE can enable GPT-4o to efficiently find solutions to challenging tasks in the BabyAI text and visual environments. In the text domain, IGE does so with orders of magnitude fewer online steps than prior RL-trained baselines (GLAM, Carta et al. (2023)). Task types are in order of difficulty. As tasks become more difficult, the performance gap of IGE vs. the LLM baselines grows. We show the mean and 95% bootstrap confidence interval (Zoubir & Iskandler, 2007) over 25 seeds per environment type. *Here, and elsewhere, confidence intervals are obtained by bootstrapped resampling 10,000 times.*

blocking objects as well as finding keys to open doors. We consider 5 different task families of increasing difficulty: "go to", "pick up", "pick up then go to", "open door", and "put next to", which are described fully in Appendix B.2.

We omit the Reflexion baseline in this environment due to the high cost of querying GPT-4 with 128-step episodes in the context. Due to the complexity of this environment, we use *chain-of-thought prompting in all three components* of IGE. This allows the FM to deliberate on the state of the game before making decisions. IGE can find solutions to these problems with only a tiny budget of 250 environment steps per task (divided into rollouts of 10 exploratory actions each) and visualize the final performance in Figure 3 (top). In the text variant, IGE and ReAct vastly outperform the prior RL-trained language model approach, GLAM (Carta et al., 2023), with orders of magnitude fewer samples (GLAM used 1.5M online steps) and requiring no training whatsoever. IGE achieves the best or close to the best performance in every task. The gap between IGE and the second-best method grows with task difficulty, with a statistically significant 36% improvement ($\chi^2$ test, $p < 0.05$) on "put next to". In the visual variant, IGE also strongly beats the visual agent baselines and displays strong performance across the board, enabling GPT-4o to successfully find solutions in a majority of cases. The results provide strong evidence that IGE extends to any modality representable by a multimodal foundation model.

## 4.3 TEXTWORLD

Finally, we show IGE's ability to tackle tasks requiring long-horizon memory and planning, exploration, and commonsense in TextWorld (Côté et al., 2018), a classic text-based agent benchmark.

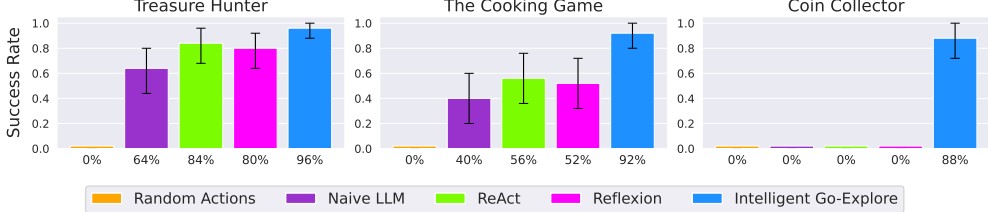

Figure 4: IGE outperforms state-of-the-art FM agents in three challenging text games in TextWorld. These results illustrate the powerful capabilities of planning, commonsense reasoning, and exploration of IGE (Section 4.3). Notably, in the Coin Collector game where hard exploration is required, we observe BFS-like search behavior emerge in IGE, enabling it to find the most efficient solution where all other approaches exhaust the environment horizon. We show the mean and 95% bootstrap confidence interval over 25 seeds for each game.

Table 2: We rigorously ablate the design choices in IGE and analyze the importance of incorporating FMs at each stage of the algorithm. Additionally, we compare IGE to Classic Go-Explore (Ecoffet et al., 2021b), which relies on manually defined heuristics at each stage. Game of 24 performance is taken at 150 environment steps, over 100 evaluation seeds. BabyAI-Text performance is taken at 250 environment steps, over 25 evaluation seeds. TextWorld performance is taken at 240 environment steps, over 25 evaluation seeds. 'Standard' is mirrored from Section 4. We show the mean and 95% bootstrap confidence interval for the success rate.

| Ablation Variants | Success Rate (%) | | |
| --- | --- | --- | --- |
| | Game of 24 | BabyAI (PN) | TextWorld (CG) |
| IGE | **100 $\pm$ 0.0** | **84 $\pm$ 14** | **92 $\pm$ 10** |
| ✗ Intelligent action selection | 68 $\pm$ 9.0 | 24 $\pm$ 16 | 0 $\pm$ 0 |
| ✗ Intelligent state selection | 96 $\pm$ 3.5 | 48 $\pm$ 20 | 76 $\pm$ 16 |
| ✗ Intelligent archive filtering | 93 $\pm$ 5.0 | 64 $\pm$ 20 | 64 $\pm$ 20 |
| ✗ All 3 above | 61 $\pm$ 9.5 | 4 $\pm$ 6 | 0 $\pm$ 0 |
| ✗ State-conditional action history | 33 $\pm$ 9.0 | 72 $\pm$ 16 | 72 $\pm$ 16 |
| Classic Go-Explore (Visitation Frequency) | 38 $\pm$ 22.0 | 4 $\pm$ 6 | 0 $\pm$ 0 |

We consider three challenging games in TextWorld: Treasure Hunter, The Cooking Game, and Coin Collector. In each game, the agent needs to complete the task while navigating a maze of different rooms, while only seeing the current room's description in text. The agent interacts with the world using free-form natural language commands, such as "go east" or "cook potato with oven." We set each game to hard difficulty; details on game customizations are provided in Appendix B.3. As in the previous section, we use chain-of-thought prompting in all three components of IGE. Because the state archive in this environment grows significantly, we implement rejection-based archive filtering, which we describe in Appendix D.2.

We present success rates achieved on the three games using IGE and the baselines in Figure 4. We observe that IGE outperforms all other baselines, with a statistically significant ($\chi^2$ test, $p < 0.05$) gap between IGE and the second-best method in the harder Cooking Game and Coin Collector. In The Cooking Game, IGE outperforms the next-best agent, ReAct, by a large margin of 36%, demonstrating IGE's advantage in hard-exploration problems. In Coin Collector, IGE *is the only method that can find the solution in the maze*, with all other methods completely failing. Interestingly, we observe that IGE exhibits BFS-like behavior, intelligently selecting rooms with unexplored directions and iteratively removing rooms with exhausted directions. This results in IGE almost always finding the shortest path to the target, while other methods fail to navigate the maze.

We highlight that Reflexion does not improve over ReAct in all the games we tested. Although Reflexion should in theory be an improvement over ReAct with the experience from previous attempts, it tends to decrease performance. We hypothesize that in long-horizon environments, the history becomes too long after the initial episode and prevents Reflexion from effectively utilizing knowledge from the previous episode. In contrast, IGE uses the FM to iteratively filter interesting states in the archive, which ends up *controlling the context length*. This helps IGE truly make use of the cumulative knowledge gained through exploration.

# 5 ANALYSIS

In this section, we analyze (1) the importance of FM intelligence for each of the three key components of Go-Explore, and (2) how IGE's performance improves as the FM's capabilities increase. We take a representative sample of environments from the previous section of Game of 24, Put Next To (PN) from BabyAI-Text, and The Cooking Game (CG) from TextWorld. Hyperparameters are listed in Appendix E.

**How Important is Foundation Model Intelligence at Each Step?** First, we analyze the impact of FM intelligence on each component of INTELLIGENT GO-EXPLORE. We ablate replacing state and action selection with uniform random sampling, archive filtering with saving everything to the archive, and not maintaining a state-conditional action history. We use these unintelligent choices, as *it would be very time-consuming to attempt to design the right heuristics* based on the rich text observations in Table 1. In Table 2, we observe that where the intelligence of FMs is more valuable varies by environment. Since the environment horizon is only 3 in the Game of 24, the most important factor is ensuring that the actions tried are diverse and intelligently selected. This hypothesis is confirmed: the largest performance drops occur when removing either FM action selection or the

action history. Different IGE components are most helpful in both of the longer-horizon BabyAI-Text and TextWorld environments: intelligent state selection and archive filtering make a big impact, showcasing the strength of enabling IGE to return to promising discovered states. There are smaller performance drops when removing the action history; likely because in larger environments, many more unique states are discovered, so there is less gain from preventing taking the same actions from frequently returned to states. In both environments, we also observe a drastic decrease when switching to random actions, as in classic Go-Explore. This underscores the substantial benefits IGE provides in harnessing FMs for action selection.

Next, we elucidate the need for intelligent archive filtering across all our environments. Not only does archive filtering improve performance, but it also drastically reduces the number of uninteresting states in the archive. For instance, in BabyAI-Text, we observe the archive becoming around $8\times$ larger without filtering. These metrics demonstrate IGE's innate ability to capture promising discoveries as they occur and focus attention on them, without the need for any manual heuristics. Detailed analysis and results are provided in Appendix C.1. Furthermore, we compare IGE with classic Go-Explore (GE, Ecoffet et al. (2021b)), which adopts fixed pre-defined, hand-designed, open-loop heuristics like resetting to states with probability inversely proportional to the state-visitation count. Across all 3 domains, IGE significantly outperforms classic GE, which shows that adopting FMs instead of manually designed heuristics not only saves the effort required for domain-specific design but also enhances the algorithm's exploration by leveraging the human notion of interestingness captured by FMs (Table 2, bottom). We particularly emphasize that the intelligence of FMs enables IGE to rapidly solve environments like BabyAI that have challenged the RL community in an exceptionally short number of 250 timesteps. This is in contrast to the classic GE paradigm, which used millions or even billions of environment steps (Ecoffet et al., 2021b).

**What is the Effect of Foundation Model Choice?** We further analyze the dependence of IGE on the capabilities of the underlying foundation model. We evaluated IGE using other foundation models, including Claude Sonnet 3.5 (Anthropic, 2024) and the open-source Llama-3 400B (Llama Team, 2024), on the Game of 24. The results, presented in Table 3, show that IGE maintains superior performance over the baselines across different foundation models, consistently achieving higher success rates. For instance, even when using the weaker Llama-3 400B, IGE achieves a success rate of $98\%$, significantly outperforming the best Llama-based baseline. Notably, the performance gains of IGE are statistically significant (for all baselines with Llama-3 400B and for all except Reflexion on Claude Sonnet 3.5, $p < 0.05$). This demonstrates that IGE is not dependent on a specific model and can be expected to perform well with other, possibly future, foundation models with higher capabilities. Additional experiments on prompt robustness are provided in Appendix C.2.

Table 3: IGE consistently outperforms other LLM agent baselines with a diverse range of foundation models on the Game of 24. We use the same evaluation setting as Table 2 and show the mean and 95% bootstrap confidence interval for the success rate (%).

| Model | Naïve LLM | ReAct | Reflexion | IGE |
|---|---|---|---|---|
| GPT-4 | $70 \pm 12$ | $82 \pm 11$ | $83 \pm 10$ | $\mathbf{100 \pm 0}$ |
| Llama-3 400B | $44 \pm 14$ | $68 \pm 13$ | $54 \pm 14$ | $\mathbf{98 \pm 3}$ |
| Claude Sonnet 3.5 | $24 \pm 12$ | $58 \pm 14$ | $80 \pm 11$ | $\mathbf{86 \pm 9}$ |

## 6 RELATED WORK

*Due to space constraints, extra related work discussions are in Appendix F.*

**FM Agents.** One of the key strengths of IGE is that it is agnostic to the precise agent formulation and thus strictly additive on top of a wide variety of strategies. A common strategy is chain-of-thought-based methods (Yao et al., 2023b; Hu & Clune, 2024), which prompts the FM to output a set of reasoning steps before the answer. We integrate this into the FM guidance in our experiments in Sections 4.2 and 4.3. Reflexion (Shinn et al., 2023) enables an agent to improve over multiple episodes by asking it to reflect on the previous attempted episode and learn from its mistakes. However, we show this can break down in tasks with long horizons, while IGE proposes a more efficient way to filter out the vast majority of uninteresting interactions. Another set of agent frameworks that are related to the idea of exploring diverse solution paths via state-connectivity is Tree of Thoughts (ToT, Yao et al. (2023a)) and Graph of Thoughts (GoT, Besta et al. (2024a)). The primary difference between methods like ToT and IGE is that ToT builds up a tree of abstract thoughts generated by the

language model and could be hallucinated, whereas IGE's archive is grounded in real states directly copied from an RL environment. Although ToT's thoughts are suitable for the types of problems it investigates, these abstract plans could easily break down in RL environments with long horizons. In contrast, IGE scales gracefully with long horizons by archiving real states and returning to the state for exploration. Additionally, ToT is based on DFS and BFS over the tree and is thus limited to these search strategies over the problem. On the other hand, IGE can intelligently select hybrid strategies, for example, going broad at the start (more BFS-like) and then honing down on a promising path (more DFS-like) later on. This is particularly important for long-horizon tasks with larger state spaces, as we show in Sections 4.2 and 4.3.

Closely related to exploration, FM agents have also begun to see use in search-based tasks. Stream of Search (Gandhi et al., 2024) considers a similar mathematical reasoning task to the Game of 24 and seeks to initially clone the actions of graph search algorithms, then use RL to self-improve. In contrast, IGE already greatly outperforms classic graph search, and an exciting future direction could be to first clone the exploratory behavior of IGE and then use that as a basis for self-improvement. Lehnert et al. (2024) analogously train a language model to mimic the $A^*$ algorithm. Finally, Krishnamurthy et al. (2024) also consider bootstrapping exploration with an externally summarized action-history in bandit problems; our focus is more on the detection of interesting states.

**Go-Explore.** The original Go-Explore (Ecoffet et al., 2021a;b) framework enabled superhuman performance in a variety of hard-exploration problems, including applications as diverse as automated game testing (Lu et al., 2024). Gallouédec & Dellandréa (2023) propose Latent Go-Explore which similarly aims to address the difficulty of designing exploration heuristics by automatically learning a latent representation and sampling states with a low latent density. However, this requires periodic retraining and could easily miss out on rare discoveries. HuGE (Torne Villasevil et al., 2023) guides Go-Explore with humans in the loop by asking for pair-wise feedback on which goal to select. On the other hand, we replace humans with intelligent FM guidance at all components of Go-Explore.

## 7 CONCLUSION AND LIMITATIONS

In this paper, we demonstrate a new approach to robust exploration in complex environments, INTELLIGENT GO-EXPLORE, reimagining Go-Explore in the era of giant foundation models. We show that IGE can drive exploration for a diverse set of FM agents, including few-shot and chain-of-thought prompting, across a variety of modalities including challenging text and vision-based games. Extending IGE to even more modalities could unlock applications as wide as scientific discovery in synthetic biology (designing novel drugs or proteins) or material science. Recent advancements in multimodal foundation models, such as RT-2 (Brohan et al., 2023) and RFM-1 (Covariant AI, 2024), have shown the potential of FMs to handle various modalities, including text, images, videos, and (continuous) numerical sensor readings. Since these models have already tokenized the state-action spaces of general environments and shown the ability of the model to generate actions for any state, it should be possible to then ask the FM to judge the interestingness of any given state compared to prior states observed, which is the only additional requirement of IGE. A further interesting domain is the (hitherto unsolved by intelligent agents) vast dungeon crawler, NetHack (Küttler et al., 2020). Küttler et al. (2020) noted that classic Go-Explore's heuristics "will likely not work for large symbolic and procedurally generated environments." IGE represents a sharp departure from these limitations by replacing hard-coded and inflexible exploration heuristics with the dynamic intelligence of giant foundation models.

There remain exciting opportunities to improve IGE's capabilities to explore vast state spaces. For example, we currently recall and compare against the entire archive whenever we discover a new state. This could be made much more efficient by using techniques like retrieval augmented generation (Lewis et al., 2020) and only comparing to the closest previously discovered states. As we consider IGE for real-world settings, we should take steps to ensure the responsible deployment of FMs (Bommasani et al., 2021). Our approach opens up the road to **safe and interpretable exploration**: through careful prompt engineering or techniques like constitutional AI (Bai et al., 2022), we could steer the agent away from unsafe behaviors. Furthermore, if we ask or train the FM to explain its choices in each part of IGE, we could gain insight into its rationale for exploring particular paths through an environment (Wei et al., 2022; Hu & Clune, 2024); improving safety, interpretability, and perhaps one day even our own understanding of how best to explore.

ACKNOWLEDGMENTS

This work was supported by the Vector Institute, the Canada CIFAR AI Chairs program, grants from Schmidt Futures and Open Philanthropy, an NSERC Discovery Grant, and a generous donation from Rafael Cosman. We thank Aaron Dharna, Ben Norman, and Jenny Zhang from our lab at the University of British Columbia for insightful discussions and feedback on early drafts of this work.

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

# SUPPLEMENTARY MATERIAL

## TABLE OF CONTENTS

## A ALGORITHM PSEUDOCODE

We provide full pseudocode for INTELLIGENT GO-EXPLORE in Algorithm 1. This complements the discussion in Section 3.

---

**Algorithm 1** INTELLIGENT GO-EXPLORE

---

1: **Hyperparameters:** no. state expansions $N_{\text{state}}$, no. exploratory actions $N_{\text{action}}$, foundation model $\mathcal{M}$
2: **Initialize:** archive of states $\mathcal{S}_{\text{archive}} = \emptyset$, state-conditional action history $\mathcal{A}(\cdot) = \emptyset$
3: $\mathcal{S}_{\text{archive}} \leftarrow \mathcal{S}_{\text{archive}} \cup \{s_0\}$ ▷ Add initial state to archive
4: **for** $i = 1, \ldots, N_{\text{state}}$ **do**
5:      Query $\mathcal{M}$ for the next state $s_{i,1}$ from $\mathcal{S}_{\text{archive}}$ ▷ See Section 3.1
6:      **for** $j = 1, \ldots, N_{\text{action}}$ **do**
7:          Query $\mathcal{M}$ for the next action $a_{i,j}$ from $s_{i,j}$ conditional on $\mathcal{A}(s_{i,j})$ ▷ See Section 3.2
8:          $s_{i,j+1} \sim P(s_{i,j}, a_{i,j}), \mathcal{A}(s_{i,j}) \leftarrow \mathcal{A}(s_{i,j}) \cup \{a_{i,j}\}$ ▷ Take action and update history
9:          **if** $\mathcal{M}$ determines that $s_{i,j+1}$ is interesting w.r.t. $\mathcal{S}_{\text{archive}}$ **then** ▷ See Section 3.3
10:              $\mathcal{S}_{\text{archive}} \leftarrow \mathcal{S}_{\text{archive}} \cup \{s_{i,j+1}\}$
11:          **end if**
12:      **end for**
13: **end for**
14: Return best discovered trajectory

---

## B FURTHER DETAILS ON ENVIRONMENTS

We provide further details for each of the environments used in the empirical evaluation in Section 4, and the environment-specific descriptions appended to the system prompts. Each environment description may include high-level information about the task and a description of the action space.

### B.1 GAME OF 24

We use the environment and set of evaluation tasks from `https://github.com/princeton-nlp/tree-of-thought-llm` which is released under the MIT License. We include the environment-specific prompt that is appended to the system prompt in Section 3 below. The system prompt contains examples of correct reasoning paths on different problems (few-shot prompting).

---

**Environment Description.**

You are given 4 numbers and must use basic arithmetic operations (+ - * /) to obtain 24. At each step, you are only allowed to choose two of the remaining numbers to obtain a new number. A correct answer is one that uses each input exactly once and no other numbers. Reaching 24 before the last step does not count as a correct answer. Follow the convention that division is integer division, and never by zero.
Some examples of correct reasoning traces are as follows:
Initial state: (4 4 6 8)
Steps:
4 + 8 = 12. Next: (4 6 12)
6 - 4 = 2. Next: (2 12)
2 * 12 = 24. Next: (24)
Answer: (6 - 4) * (4 + 8) = 24
Initial state: (2 9 10 12)
Steps:
12 * 2 = 24. Next: (9 10 24)
10 - 9 = 1. Next: (1 24)
24 * 1 = 24. Next: (24)
Answer: (12 * 2) * (10 - 9) = 24
Initial state: (4 9 10 13)
Steps:
13 - 10 = 3. Next: (3 4 9)
9 - 3 = 6. Next: (4 6)
4 * 6 = 24. Next: (24)

---

> Answer: 4 * (9 - (13 - 10)) = 24

The action space at each step is all the valid arithmetic operations, presented in an analogous way as the 'propose' step in Yao et al. (2023a).

## B.2    BABYAI

The BabyAI (Carta et al., 2023) environment comes with five task types, which we list here and visualize in order in Figure 5:

- Go to <object>, a simple navigation task that requires reasoning abilities to choose the right plan given the object's position;
- Pick up <object>, a reasoning task that combines navigation tasks;
- Pick up <object A> then go to <object B> and Go to <object B> after pickup <object A>, both serving to test reasoning abilities on temporal sequences;
- Unlock <door>, a task that includes inferring that a key is needed to unlock the door, finding the right key (i.e., the one colored as the door), and eventually using the toggle action with the key on the door;
- Put <object A> next to <object B>, which requires first reaching <object A>, picking it up, reaching <object B> and finally dropping <object A> next to <object B>.



Figure 5:   We visualize the 5 types of tasks that BabyAI consists of for our evaluation in Section 4.2. IGE receives only partial text-based observations corresponding to the view in the figure.

We use the codebase from `https://github.com/flowersteam/Grounding_LLMs_with_online_RL` which is released under the MIT License. The action space is discrete and composed of 6 possible actions: turn left, turn right, go forward, pick up, drop, and toggle. The 'go to' and 'pick up' tasks have a shorter environment horizon of $H = 64$, whereas the rest have a horizon of $H = 128$. We include the environment-specific prompt that is appended to the system prompt in Section 3 below.

> **Environment Description (Text).**
>
> You are an agent in an 8x8 partially-observable 2D text-based environment. You see the 6x6 grid in front of you, and can face north, south, east, or west. The possible actions are turn left, turn right, go forward, pick up, drop, and toggle. At each turn, you will receive a history of the last observations and actions. Your aim is to complete the task described in the goal. Each tile in the grid can only contain at most one object. Objects cannot be crossed, and may need to be bypassed or moved. You can only move onto an empty tile or on a tile containing an open door. You can only hold one object at a time, using pick up when they are one step in front. Objects are dropped one tile in front and cannot be dropped when there is another object in front. Doors are unlocked with keys of the same color using the toggle action. Actions are deterministic, do not repeat actions if they have no effect. You have H steps to complete the task.

> **Environment Description (Visual).**
>
> You are an agent in an 8x8 partially-observable 2D image-based environment. You see the 6x6 grid in front of you, and can face north, south, east, or west. The possible actions are turn left, turn right, go forward, pick up, drop, and toggle. At each turn, you will receive a history of the last observations

> and actions. The observations will be presented as a sequence of images. Your aim is to complete the task described in the goal. Each tile in the grid can only contain at most one object. Objects cannot be crossed, and may need to be bypassed or moved. You can only move onto an empty tile or on a tile containing an open door. You can only hold one object at a time, using pick up when they are one step in front. Objects are dropped one tile in front and cannot be dropped when there is another object in front. Doors are unlocked with keys of the same color using the toggle action. Actions are deterministic, do not repeat actions if they have no effect. You have H steps to complete the task.

## B.3   TEXTWORLD

We evaluate IGE on 'Treasure Hunter', 'The Cooking Game', and 'Coin Collector' from the TextWorld (Côté et al., 2018) domain. We use the environment code from `https://github.com/microsoft/TextWorld` which is released under the MIT License.

### B.3.1   TREASURE HUNTER

For Treasure Hunter, we set the 'level' option to the maximum value of 30, resulting in a maze with 20 rooms. Locked doors and containers are added, which may need to be unlocked and opened to find the target object. To further increase the difficulty, we remove the solution description from the original game and filter out tasks that can be completed with 20 steps in the optimal solution. We include the environment-specific prompt that is appended to the system prompt in Section 3 below.

> **Environment Description for Treasure Hunter.**
>
> You are an agent playing TextWorld, a text-based adventure game where you are in a randomly generated maze and must find a specific object. You need to explore different rooms to find the target object.
> Here are the available commands: look: describe the current room. goal: print the goal of this game inventory: print the player's inventory go <dir>: move the player north, east, south, or west. You can only go in the direction indicated with an exit or a door. open ...: open a door or a container. You need to open a closed door before you want to go through it. drop ...: drop an object on the floor take ...: take an object that is visible. Make sure the object is visible to take. put ... on ...: place an object on a supporter take ... from ...: take an object from a container or a supporter insert ... into ...: place an object into a container unlock ... with ...: unlock a door or a container with a key. You need to unlock a locked door with a matched key in your inventory before you want to open it.
> - The target object might be located in a closed or locked container. - The adjective is useful for determining whether the key is matched with the lock (e.g., non-euclidean keycard is matched with non-euclidean safe). Make sure it is matched to unlock! - The key required to unlock the door may be in another room or locked inside a container. - Take the key whenever you can. - After unlocking a locked door or container, it will remain closed. You will then need to open it.
> You have 40 steps to complete the task. Restarting is forbidden.

### B.3.2   THE COOKING GAME

In The Cooking Game, we set the number of ingredients to a maximum of 5 and the number of rooms to 13. We enable all challenging additional options: doors need to be opened, food must be processed (e.g., cut, diced, chopped with a knife), and cooked (e.g., grilled with a BBQ, fried on a stove, roasted in an oven). We include the environment-specific prompt that is appended to the system prompt in Section 3 below.

> **Environment Description for The Cooking Game.**
>
> You are an agent playing TextWorld, a text-based adventure game where you navigate through different rooms, interact with objects, and solve puzzles. Your goal is to first find the recipe, find and prepare food according to the recipe, and finally prepare and eat the meal.
> Here are the available commands: look: describe the current room goal: print the goal of this game inventory: print player's inventory go <dir>: move the player north, east, south or west. You can only go to directions indicated with an exit or a door. examine ...: examine something more closely eat ...: eat edible food open ...: open a door or a container. You need to open a closed door before you can go through it. drop ...: drop an object onto the floor take ...: take an object that is visible put ... on ...: place an object on a supporter take ... from ...: take an object from a container or a supporter insert ... into ...:

place an object into a container lock ... with ...: lock a door or a container with a key unlock ... with ...: unlock a door or a container with a key cook ... with ...: cook cookable food with something providing heat slice ... with ...: slice cuttable food with something sharp chop ... with ...: chop cuttable food with something sharp dice ... with ...: dice cuttable food with something sharp prepare meal: combine ingredients from inventory into a meal. You can only prepare meals in the Kitchen.
- You can examine the cookbook to see the recipe when it is visible. - The BBQ is for grilling things, the stove is for frying things, the oven is for roasting things. Cooking ingredients in the wrong way will lead to a failure of the game. - Once you have got processed ingredients and the appropriate cooking tool ready, cook all of them according to the recipe. - There are two conditions to correctly cook something (grill/fry/roast): a) the ingredient you want to cook is in your inventory and b) there is a suitable cooking tool in the room, and then use 'cook ... with ...' command. - When you need to chop/slice/dice ingredients, you need to take the knife and the ingredient in your inventory and then 'slice/chop/dice ... with knife' - Make sure to first process the food (chop/slice/dice) before you try to cook them. - When you have all the ingredients (that got processed or cooked according to the menu), you can 'prepare meal' in the kitchen and then 'eat meal' to win the game. - The ingredients should EXACTLY match the color in the recipe, but if the recipe doesn't specify color, any color would be fine. When you 'take ... with ...', use the EXACT name you see. - You don't need to examine the container/supporter (e.g., toolbox) when it says something like "there isn't a thing on it"/"has nothing on it"
You have 80 steps to complete the task. Restarting is forbidden.

We show a successful example trajectory found by IGE below, from our evaluation in Section 4.3.

**Example trajectory found by IGE in The Cooking Game.**

You are hungry! Let's cook a delicious meal. Check the cookbook in the kitchen for the recipe. Once done, enjoy your meal!
-= Backyard =- You find yourself in a backyard. A normal kind of place. You can see a patio table. But oh no! there's nothing on this piece of garbage. This always happens! You make out a patio chair. But oh no! there's nothing on this piece of trash. You can make out a BBQ, so there's that. There is a closed barn door leading north. There is a closed sliding patio door leading west. There is an exit to the east. Don't worry, there is no door.
> open sliding patio door
You open sliding patio door.
> go west
-= Kitchen =- You find yourself in a kitchen. An ordinary kind of place. The room is well lit. You can make out a fridge. You hear a noise behind you and spin around, but you can't see anything other than an oven. Were you looking for a table? Because look over there, it's a table. On the table you see a cookbook. You can't wait to tell the folks at home about this! Look out! It's a- oh, never mind, it's just a counter. Now why would someone leave that there? You see a red apple, a raw purple potato, a red onion, a yellow apple and a knife on the counter. You can see a stove. You shudder, but continue examining the stove. Unfortunately, there isn't a thing on it. Hm. Oh well There is an open sliding patio door leading east. There is a closed plain door leading north. You don't like doors? Why not try going west, that entranceway is not blocked by one.
> examine cookbook
You open the copy of "Cooking: A Modern Approach (3rd Ed.)" and start reading:
Recipe #1 ——— Gather all following ingredients and follow the directions to prepare this tasty meal. Ingredients: block of cheese carrot orange bell pepper red apple tomato Directions: chop the block of cheese fry the block of cheese dice the carrot grill the carrot chop the orange bell pepper fry the orange bell pepper chop the red apple grill the red apple slice the tomato grill the tomato prepare meal
> take red apple
You take the red apple from the counter.
Your score has just gone up by one point.
> open fridge
You open the fridge, revealing a block of cheese, a carrot, a white onion and a yellow bell pepper.
> take block of cheese
You take the block of cheese from the fridge.
Your score has just gone up by one point.
> take carrot
You take the carrot from the fridge.
Your score has just gone up by one point.
> examine counter
The counter is solid.

On the counter are a raw purple potato, a red onion, a yellow apple and a knife.
> go west
-= Livingroom =- You arrive in a livingroom...
... (skip 30 steps)
> cook tomato with BBQ
You grilled the tomato.
Your score has just gone up by one point.
> go west
-= Kitchen =- You find yourself in a kitchen...
> prepare meal
Adding the meal to your inventory.
Your score has just gone up by one point.
> eat meal
You eat the meal. Not bad.
Your score has just gone up by one point.
*** The End ***
You scored 17 out of a possible 17, in 44 turns.

### B.3.3 COIN COLLECTOR

In Coin Collector, we set the number of rooms to 40 and allow distractor rooms to be added along the way. Similar to Treasure Hunter, we remove the solution description from the original game, and the optimal path from the agent's starting point to the target is set to 20 steps. We include the environment-specific prompt that is appended to the system prompt in Section 3 below.

**Environment Description for Coin Collector.**

You are an agent playing TextWorld, a text-based adventure game where you are in a randomly generated maze and must find the coin. You need to explore different rooms to find the target object.
Here are the available commands: goal: print the goal of this game go <dir>: move the player north, east, south, or west. You can only go in the direction indicated with something like an exit or a door. take coin: win the game by 'take coin' if you see the coin in the room
The only action you can do is 'go <dir>' to explore the maze and 'take coin' when you see the coin in the room.
You have 25 steps to complete the task. Restarting is forbidden.

## C ADDITIONAL EXPERIMENTS AND ANALYSIS

### C.1 ARCHIVE SIZE ANALYSIS

We provide the detailed analysis of the impact of intelligent archive filtering on the size of the archive across different environments. As shown in Table 4, intelligent filtering in IGE can drastically reduce the size of the archive and help the algorithm focus on the most interesting states. Without intelligent filtering in BabyAI-Text and TextWorld, the archive becomes over $5\times$ larger.

Table 4: We show that intelligent filtering in IGE can drastically reduce the size of the archive, and help the algorithm focus on the most interesting states. We use the same evaluation setting as Table 2 and show the mean and 95% bootstrap confidence intervals.

| Archive Filtering | Number of States | | |
| | Game of 24 | BabyAI (PN) | TextWorld (CG) |
|---|---|---|---|
| No Filter | $18.5 \pm 3.2$ | $203.5 \pm 56.7$ | $22.4 \pm 15.3$ |
| With Filter | $\mathbf{15.6 \pm 2.3}$ | $\mathbf{25.5 \pm 5.2}$ | $\mathbf{4.4 \pm 2.8}$ |

These results highlight that, across all evaluated environments, intelligent archive filtering not only enhances performance but also maintains a manageable archive size, which is crucial for scalability.

## C.2 Prompt Robustness Experiments

We investigated the sensitivity of IGE to prompt variations to assess its robustness to prompt engineering. Specifically, we conducted experiments where we removed any hints or domain-specific instructions on how to solve the tasks, keeping the prompts purely factually descriptive about the environment. For example, in:

- Game of 24
    - We removed all few-shot prompting, so that the agent has no examples. I.e., all text from "Some examples of correct reasoning traces are as follows:"
- BabyAI
    - We removed "do not repeat actions if they have no effect"
- TextWorld (Cooking Game)
    - We removed "The ingredients should EXACTLY match the color in the recipe, but if the recipe doesn't specify color, any color would be fine. When you 'take ... with ...', use the EXACT name you see."
    - We removed "You don't need to examine the container/supporter (e.g., toolbox) when it says something like "there isn't a thing on it"/"has nothing on it" "

We refer to these as domain-general prompts, as they contain only basic information about the environment without any guidance on how to approach the tasks.

The evaluation settings remained identical to those in our main experiments (see Section 4), ensuring a fair comparison. The results, shown in Table 5, indicate that even with minimal prompts, IGE maintains a significant performance advantage over the baselines across all environments. For instance, in the Game of 24, IGE achieves a success rate of 96% with domain-general prompts, compared to 100% with the original prompts, and still significantly outperforms the baselines. Similar trends are observed in the BabyAI "Put Next To" task and the TextWorld "Cooking Game". The performance drop of IGE when using domain-general prompts is modest compared to the baselines, demonstrating that IGE is robust to prompt variations and does not rely heavily on domain-specific prompt engineering.

These findings suggest that IGE can be applied effectively without extensive prompt tuning, making it flexible and generalizable to new domains. The foundation model's internalized notions of interestingness and problem-solving abilities enable IGE to guide exploration even with minimal guidance, highlighting its potential for widespread applicability.

Table 5: IGE Performance with Domain-General Prompts. We use the same evaluation setting as Table 2 and show the mean and 95% bootstrap confidence interval for the success rate.

| Environment | Model | Naïve LLM | ReAct | Reflexion | IGE |
|---|---|---|---|---|---|
| Game of 24 | Original (GPT-4) | $70 \pm 14$ | $82 \pm 12$ | $83 \pm 11$ | $\mathbf{100 \pm 0}$ |
| | Domain-General Prompt | $35 \pm 13$ | $71 \pm 13$ | $71 \pm 13$ | $\mathbf{96 \pm 4}$ |
| BabyAI "Put Next To" | Original (GPT-4) | $12 \pm 12$ | $48 \pm 12$ | N/A | $\mathbf{84 \pm 8}$ |
| | Domain-General Prompt | $8 \pm 12$ | $24 \pm 12$ | N/A | $\mathbf{68 \pm 12}$ |
| TextWorld "Cooking Game" | Original (GPT-4) | $40 \pm 16$ | $56 \pm 16$ | $52 \pm 16$ | $\mathbf{92 \pm 8}$ |
| | Domain-General Prompt | $44 \pm 16$ | $48 \pm 16$ | $48 \pm 16$ | $\mathbf{72 \pm 16}$ |

# D Further Prompt Discussion

## D.1 Extracting Choices

By default, in Section 4.1, we prompt the FM to return a JSON object containing just the numerical index of the choice. We choose this because of the ease of parsing the response and validating it lies within the correct bounds. An example prompt is displayed below.

> **JSON Choice Prompt.**
>
> Reply concisely and exactly with the following JSON format:
> {"choice": X}
> where X is the index of the desired choice.

When using chain-of-thought as in Section 4.2, we use the following prompt:

> **JSON Choice Prompt (Chain of Thought).**
>
> First, briefly reason about your plan.
> Reply concisely and exactly with the following JSON format:
> {"thought": X, "choice": Y}
> where X is your reasoning and Y is the index of the desired choice. Make sure Y is a parsable integer.

For the TextWorld environment in Section 4.3, since the action space is much larger, we ask the FM to directly output a text action that we automatically parse.

> **Decision Making Prompt.**
>
> Please briefly reason about your plan and then output the command in the format '> command'. Ensure only one command is included.

We use the regex "> (.*?)(?:-–$)" (in Perl notation) to parse the command. We note that the failure rate for both of these options is very low, less than 0.1% across our evaluation. Despite this, we include a failsafe that returns a random choice in case of an invalid output.

## D.2 REJECTION-BASED ARCHIVE FILTERING

The 'acceptance-based' archive filter in Section 3.3 iterates through every new state and asks whether each one is interestingly new and should be added to the archive. This approach can break down in larger environments where it becomes necessary to explicitly deprecate earlier discoveries that have become irrelevant, in order not to overload the archive (e.g., in Section 4.3). In this environment, we use an alternate version of the prompt which first adds all states, and then asks the foundation model to remove the uninteresting states. An example prompt is shown below.

> **Rejection-based Archive Filtering Prompt.**
>
> Current state archive:
> [State Archive]
> Remove outdated states that are no longer relevant to the task, have had all interesting explorations attempted, or have similar states in the archive that show more progress.

## D.3 IMPLEMENTATION OF STATE-CONDITIONAL ACTION HISTORY

The state-conditional action history in IGE stores the set of actions previously attempted from each archived state. For each state in the archive, we maintain a list of actions that have been tried from that state, which helps prevent the foundation model from repeating the same actions and encourages exploration of new actions. This history is provided in the context when prompting the foundation model for the next action. Since our state archive is kept at a manageable size (see Table 4), this additional memory requirement is minimal.

## D.4 HANDLING HALLUCINATIONS DURING TRAJECTORY CREATION.

Foundation models are known to sometimes produce hallucinations or invalid outputs. In IGE, the foundation model does not create entire trajectories; instead, it is queried for state/action choices from actual options, and the resulting states are saved into an archive, as in the original Go-Explore. Therefore, IGE provides robust scaffolding for foundation model agents that explicitly prevent the hallucination of impossible states. There is a concern that the LLM could pick impossible actions, but we observe that malformed inputs only consist of less than 0.1% of total actions. As discussed

in Appendix D, in these rare cases, we can simply take the simple choice of randomly sampling an action from the environment. Further work could explore additional safeguards and validation mechanisms to handle hallucinations more effectively.

# E HYPERPARAMETERS

In this section, we provide the hyperparameters for our empirical evaluation in Section 4. We list the hyperparameters for IGE in Table 6. We choose the values for exploratory rollout length based on the average number of steps needed to make 'reasonable progress' in the environment.

Table 6: IGE Sampling Parameters. TH, TCG, and CC are abbreviations for Treasure Hunter, The Cooking Game, and Coin Collector in TextWorld.

| Hyperparameter | Value(s) | | | | |
| --- | --- | --- | --- | --- | --- |
| | Game of 24 | BabyAI | TH | TCG | CC |
| No. state expansions, $N_{\text{state}}$ | 50 | 25 | 24 | 48 | 125 |
| No. exploratory actions, $N_{\text{action}}$ | 3 | 10 | 5 | 5 | 1 |

We list the sampling parameters for GPT-4 (OpenAI, 2024) passed via the OpenAI API in Table 7.

Table 7: GPT-4 Sampling Parameters

| Hyperparameter | Value | | |
| --- | --- | --- | --- |
| | Game of 24 | BabyAI | TextWorld |
| Temperature | 0.7 | 0.7 | 0.3 |
| Max new tokens | 1000 | 1000 | 1000 |
| Response format | JSON Object | JSON Object | Text |
| Version | Turbo-2024-04-09 | o-2024-05-13 | o-2024-05-13 |

We used GPT-4-Turbo for Game of 24 and GPT-4o for BabyAI and TextWorld. This was purely done to select the version of GPT-4 that was available and the cheapest at the time of running the experiments. The version of GPT-4 is consistent per environment. We use a reduced temperature for the TextWorld domain to reduce the possibility of generating malformed responses, as actions are output in free-form natural language. In our ablations in Section 5, we use the 'turbo-0125' variant of GPT-3.5.

## E.1 COST OF EXPERIMENTS

We provide the average cost per task for our algorithm per environment (the number of seeds is specified in Section 4):

Table 8: Per task API cost for IGE using GPT-4 listed in USD.

| Environment | API Cost (USD) |
| --- | --- |
| Game of 24 | 1.04 |
| BabyAI | 2.01 |
| TextWorld | 1.28 |

We note that the price per token of the 'o-2024-05-13' option is half that of 'Turbo-2024-04-09', so we could expect to achieve the same level of results on the Game of 24 with half the price. The total cost of API access required to perform the final experiments in this paper was under 2,000 USD. During development, we iterated on IGE with a smaller number of seeds, which represents a small fraction of this cost added on top.

## F MORE RELATED WORK AND FUTURE WORK

**FM-as-judge.** We employ FM guidance at all stages of IGE to drive exploration. FMs as judges (Zheng et al., 2023a; Bradley et al., 2023) have already seen use in decision-making tasks: OMNI (Zhang et al., 2024) considers FM guidance in multi-task settings to select the most promising next task to train on. However, focusing on the broader task could miss out on interesting behavior that happens at a more granular level, and thus IGE greatly expands on the integration of FM intelligence into decision-making. RL from AI Feedback (Bai et al., 2022; Lee et al., 2024; Klissarov et al., 2023) considers training RL agents using reward functions derived from FM preferences. This similarly guides agents towards preferred states but without the intelligence of FMs for action selection.

**Extension to Stochastic Environments.** Our current experiments focus on deterministic environments to clearly demonstrate the effectiveness of IGE. However, IGE can be extended to stochastic environments, similar to how the original Go-Explore (Ecoffet et al., 2021b) was extended. In the 2021 Go-Explore work, a robust goal-conditioned policy is trained via imitation learning on the trajectories found during exploration, enabling generalization to stochasticity. In the context of IGE, we can similarly collect successful trajectories and provide them in-context to the foundation model, leveraging its ability to generalize and handle stochastic outcomes. By incorporating past trajectories into the FM's context, IGE can learn to navigate stochastic transitions and maintain robust exploration strategies. This represents an exciting direction for future research.

**Relationship with Hierarchical Search and MCTS.** The Go-Explore framework, and by extension IGE, shares similarities with hierarchical search and Best-First Search (BFS), as it prioritizes exploration from the most promising states. However, IGE builds on this approach by leveraging the foundation model's intelligence to dynamically assess the interestingness and potential of states, rather than relying on fixed heuristics. This allows IGE to adaptively explore the search space in a more informed manner. In extremely hard combinatorial problems like chess and Go, integrating an effective interestingness function is crucial. If we could train or obtain such a function, combined with a solid value function similar to those used in Monte Carlo Tree Search (MCTS), IGE could potentially discover novel and interesting strategies. Exploring this integration with MCTS and developing advanced interestingness functions through FMs are promising avenues for future work.

**Comparison to GFlowNets.** GFlowNets (Bengio et al., 2023) aim to sample compositional structures proportionally to a specified reward function, which differs from IGE's goal of exploring and discovering interesting states without predefined rewards. While both involve generating states through sequential decisions, IGE focuses on leveraging the foundation model's notions of interestingness rather than sampling according to a reward distribution. Moreover, GFlowNets are not directly intended for hard-exploration or sparse-reward problems. IGE should therefore be much better in hard exploration tasks where there is little to no reward signal to guide search. We believe a detailed comparison with GFlowNets is an interesting direction for future research.

**Comparison with Exploration Methods in Games like Minecraft.** Recent works in Minecraft, such as Voyager (Wang et al., 2023) and Ghost in the Minecraft (Zhu et al., 2023b), have employed agents that act in the environment via collections of high-level code or algorithmic policies tailored to Minecraft. In contrast, IGE provides a generic method to operate in any state and action space, given a minimal description of the environment, by leveraging the intelligence of foundation models at test time. The code policies used in these Minecraft agents are indeed interesting and could potentially be integrated into an IGE-like framework, representing a promising direction for future research into more efficient exploration agents. By combining the strengths of both approaches, we could enhance the exploration capabilities of agents in complex environments.

**Potential Real-life Applications.** IGE's framework is very general, and we envision that it could be extended to many real-life problems involving exploration in complex spaces. For example, in synthetic biology, IGE could aid in discovering novel proteins or designing new drugs. In mathematics, it could assist in exploring mathematical conjectures or finding novel proofs. Recent work has begun to explore applications of LLM agents across science, such as in biology research (Laurent et al., 2024) and solving olympiad-level math problems (Trinh et al., 2024). On the practical side, LLMs have been adapted for various web-browsing and computer-based tasks (Liu et al., 2023) as useful personal assistants, many of which require exploration across long horizons (e.g., building an app

from scratch). By leveraging IGE, these agents could improve in planning and exploration in such complex tasks.

**Accelerating Open-endedness Research.** Open-endedness is often formulated as an unsupervised exploration problem, aiming to develop algorithms that can continually generate novel and diverse behaviors or solutions without predefined objectives. IGE provides a general mechanism to explore and discover a diverse set of interesting states or solutions in arbitrary environments, much like human scientific exploration. By leveraging FMs' internalized notions of interestingness, IGE can identify and pursue novel directions that may not be specified explicitly. This capability could accelerate research in open-endedness by providing a powerful tool for unsupervised exploration in complex domains.

**Robustness and Potential Biases.** Foundation models are known to have biases stemming from their training data, which could affect their performance and decision-making in certain environments. Although representation in pre-training is hard to measure even for open-weight models, we have strong evidence that IGE's success is relatively independent of the base LLM performance. For instance, in our new results in Table 3, with Claude Sonnet 3.5 on Game of 24, the naive action selection performance is low (24%), but IGE raises it significantly to 86%. The general question of bias is significant; for example, if a foundation model was biased to think that nothing "purple" is interesting, we wouldn't explore states that are "purple". We do not observe such issues in our current evaluation (indeed, we would expect them to manifest less in games and RL/control problems rather than human-centric problems), but this raises an interesting point tied to the wider literature of reducing bias in foundation models in general. Future work could investigate methods to detect and mitigate potential biases in IGE to ensure fair and unbiased exploration.

**Potential for Fine-tuning or RL Training.** Fine-tuning or additional RL training could potentially enhance foundation models' performance in specific tasks. In the context of IGE, one could envision fine-tuning the foundation model on the trajectories discovered during exploration, further improving its decision-making and ability to generalize. Our current approach shows that significant gains can already be achieved without additional training, which is advantageous for efficiency. Additionally, the trajectories discovered by IGE could be used to train traditional RL or imitation learning algorithms, providing valuable data for further improvements. Exploring these possibilities could lead to more powerful agents capable of solving even more complex tasks.

**Investigating LLMs' Notions of Interestingness.** Understanding and formalizing the foundation model's notions of interestingness is an intriguing direction for future research. Similar to how reward models are collected on human preferences and used to fine-tune LLMs (Ouyang et al., 2022), we could consider extracting interestingness preferences and fine-tuning an LLM to select more interesting states or actions. This could then lead to more efficient versions of IGE or even improvements to FMs themselves. Developing methods to quantify and interpret the interestingness judgments made by FMs could also enhance our understanding of their decision-making processes.

**Alternate Action Selection Strategies.** Random action selection and FM action selection represent two extremes in action selection strategies. Our framework is flexible, and we could consider substituting the action selection policy in IGE with any RL policy. This would enable us to keep the foundation model IGE scaffolding that prevents detachment and derailment and allow us to use more computationally efficient options to roll out trajectories. For example, integrating exploration strategies based on intrinsic motivation or curiosity-driven policies could enhance scalability. Exploring such combinations could provide a balance between performance and computational cost, extending the applicability of IGE to larger and more complex environments.

