# OpenReview forum: "Intelligent Go-Explore: Standing on the Shoulders of Giant Foundation Models"
_ICLR.cc/2025/Conference — ICLR 2025 Poster_

### Official Review · Reviewer_ZTB4 · 2024-10-18

**Soundness:** 2
**Presentation:** 3
**Contribution:** 2
**Rating:** 6
**Confidence:** 4

**Summary:**

The authors propose Intelligent Go-Explore (IGE), an extension of the original Go-Explore algorithm that replaces manually crafted heuristics with foundation models (FMs) to guide exploration. In the original Go-Explore, heuristics were required to identify 'interesting' states, whereas in IGE, this is determined by querying the foundation model, which leverages its pretrained knowledge to make decisions. Interesting states are stored in an archive, and the foundation model is queried with which state should be explored next. Additionally, instead of relying on random action sampling, IGE uses the foundation model as a decision-making agent to choose actions intelligently. The authors evaluate IGE in controlled environments (BabyAI and Game of 24), showing that it outperforms previous foundation model-based agents in these tasks.

**Strengths:**

- The integration of foundation models with the Go-Explore framework is a noteworthy contribution that can alleviate several of the inflexible inconveniences present in the original framework, such as hand-crafted heuristics and random exploration.
- The authors perform a solid empirical evaluation and compare it to several foundation model agent baselines, demonstrating significant performance gains.
- For each of the proposed additions to the original algorithm, a solid ablation study is performed where each addition is omitted and evaluated, and IGE is also compared to the original Go-Explored in this experiment.
- The authors do not limit their approach to text-based representations and demonstrate that IGE can also be applied with visual observations.

**Weaknesses:**

- One of the main contributions of the paper is alleviating the original Go-Explore from manual handcrafting of domain-specific heuristics. However, by replacing this part of the process with foundation models, quite extensive and domain-specific prompt-engineering is required instead (as shown in Appendix B).
- Given that the environments used in the paper are relatively controlled and intuitive, and the foundation models were provided with detailed explanations of the environment dynamics and objectives, it is not clear how well the decision-making of the foundation models would work in more complex (and actual hard-exploration) environments. For instance, the Game of 24 is a relatively simple mathematical puzzle, where the exploration space is highly constrained and the rules are clear and deterministic. As such, it doesn’t pose a true hard-exploration challenge, which is what Go-Explore and IGE are meant to address.
- Building on the previous weakness, the original Go-Explore algorithm was demonstrated across the Atari suite, including hard-exploration games like Montezuma's Revenge. In this IGE paper, the authors compared Go-Explore in specific environments where IGE performs well, but it would have perhaps made more sense to evaluate it in the Atari suite, where we already know Go-Explore performs well. It is not clear that a foundation model would be able to deal with the complexity of Atari games both in judging interesting states and in decision-making. Given that the main claim of the paper is improving upon the original framework, a comparison on that suite would have provided a more direct measure.
- Although the authors touch upon this briefly, IGE appears computationally expensive, as the foundation model is presented with the entire archive of states to select a state, filters states, and is also queried for every individual action. While this may be feasible in the small environments used in the paper, it’s unclear how scalable the approach would be to larger environments or longer-horizon tasks.
- Following this, according to the ablation study, the primary driver of IGE’s performance is the action selection guided by the foundation model, compared to random action selection. However, foundation models tend to only perform well in relatively intuitive and deterministic domains where decisions are relatively straightforward. In more complex or large-scale environments, a reinforcement learning exploration policy will likely be necessary. This raises an important question: if the primary performance gains from using IGE come from action selection, and these foundation models struggle in more complex environments, are the gains from using foundation models for state selection and filtering substantial enough to justify its use? The ablation study shows that these gains are minor (yet introduce significant computational overhead), calling into question the overall value of IGE when considering scalability and generality.
- The paper occasionally comes across as over-optimistic with respect to the proposed algorithm and e.g. speculates that its solutions can easily be used for downstream RL tasks or infinite in-context learning with no evidence supporting this. A more grounded discussion of IGE's limitations would benefit the paper.

**Questions:**

- Could you provide more details on the sensitivity of the prompts used with IGE? For example, in the BabyAI environment, you explicitly prompt the model with instructions such as 'do not repeat actions if they have no effect.' How much does IGE's performance depend on these kinds of statements?
- Given that the action selection of the foundation model has the most significant impact on IGE's performance, have you explored other exploration approaches? It seems that random actions and foundation model-driven actions represent two extremes. Have you considered comparing IGE to a relatively simple information-seeking policy or other exploration strategies that could computationally be much cheaper to use?

---

> ### Author Response · Authors · 2024-11-22
> **Response to ZTB4 (Part 1/2)**
>
> We thank the reviewer for recognizing the contributions of our work, noting that "the integration of foundation models with the Go-Explore framework is a noteworthy contribution that can alleviate several of the inflexible inconveniences present in the original framework, such as hand-crafted heuristics and random exploration." We appreciate that you find our empirical evaluation solid and value our ablation studies.
>
> **Issue 1 and Question 1: Addressing Prompt Engineering Concerns**
>
> Thank you for raising this concern, we conducted experiments to show that IGE remains effective even with no hints on how to solve the task and just relying on purely factual environment descriptions. For example, as you suggested, in the BabyAI environment, we removed statements like "do not repeat actions if they have no effect". The results, shown in Table 1 below, indicate that even with minimal prompts, IGE **strongly outperforms baselines (statistically significant, p < 0.05)** with a domain-general prompt (save for a description of that environment). The evaluation setting is identical to that of Table 2 of our paper. Even with these minimal prompts, IGE outperforms baselines, indicating that extensive prompt engineering is not necessary. This shows that IGE is independent to any prompt engineering and instead is strictly additive on top of any foundation model agent architecture. We have updated the revised paper to reflect this finding in Appendix C.2.
>
>    **Table 1: IGE Performance with Domain-General Prompts**
>
>    **Game of 24**
>
> | Model                  | Naive LLM         | ReAct            | Reflexion        | IGE               |
> |------------------------|-------------------|------------------|------------------|-------------------|
> | Original (GPT-4)       | 0.70 ± 0.14       | 0.82 ± 0.12      | 0.83 ± 0.11      | 1.00 ± 0.00       |
> | Domain-General Prompt (GPT-4) | 0.35 ± 0.13 | 0.71 ± 0.13      | 0.71 ± 0.13      | 0.96 ± 0.04       |
>
>    **BabyAI "Put Next To"**
>
> | Model                  | Naive LLM         | ReAct            | IGE               |
> |------------------------|-------------------|------------------|-------------------|
> | Original (GPT-4)       | 0.12 ± 0.12       | 0.48 ± 0.12      | 0.84 ± 0.08       |
> | Domain-General Prompt (GPT-4) | 0.08 ± 0.12       | 0.24 ± 0.12      | 0.68 ± 0.12       |
>
>    **TextWorld "Cooking Game"**
>
> | Model                  | Naive LLM         | ReAct            | Reflexion        | IGE               |
> |------------------------|-------------------|------------------|------------------|-------------------|
> | Original (GPT-4)       | 0.40 ± 0.16       | 0.56 ± 0.16      | 0.52 ± 0.16      | 0.92 ± 0.08       |
> | Domain-General Prompt (GPT-4) | 0.44 ± 0.16       | 0.48 ± 0.16      | 0.48 ± 0.16      | 0.72 ± 0.16       |
>
>
> **Issues 2, 4, and 5: Applicability to More Complex Environments**
>
> While the reviewer is right that the environments in the paper considered are “relatively controlled and intuitive”, they still pose a huge challenge for current RL algorithms. For example, we have a direct comparison to SOTA RL algorithms like the language model-based GLAM [Carta et al., ICML 2023] on the BabyAI environment and find that it gets next to 0 performance on the harder “open door” and “put next to” environments. We agree that testing IGE on Atari would be extremely interesting future work, but we believe it is out of scope for this paper which introduces a novel powerful algorithm and already shows strong results on environments that have eluded RL papers that have been accepted to prior conferences. Additionally, we believe that it is often best for algorithms to be introduced with problems that are simple and intuitive, as they help reveal the principles that make the new algorithm work well, and this is a common approach in machine learning papers.
>
> Furthermore, we respectfully disagree that FMs are known to struggle in more complex environments. Indeed, we have already observed multimodal foundation models becoming an essential tool in extremely promising RL research, including being capable of playing classic video games like Doom [de Wynter, 2024] zero-shot without training, generalizing to a large number of (and even held out) 3D video games [Baker et al., 2021, Wang et al., 2023, SIMA Team, 2024], and succeeding on extremely difficult real-world robotics tasks, including “any-to-any” models such as RT-2 [Brohan et al., 2023] and RFM-1 [Covariant AI, 2024] that can reason over “text, images, videos, robot actions, and numerical sensor readings”. Overall, combining FMs + RL is one of the most active, promising directions in RL, and is in our view at least worthy of research to study its potential. We have added a discussion on this in the conclusion of our paper.

---

> ### Author Response · Authors · 2024-11-22
> **Response to ZTB4 (Part 2/2)**
>
> Since our algorithm is strictly additive on top of the capabilities of any FM agent, there is little reason to doubt that IGE’s capabilities will improve in tandem with the ever-growing capabilities of modern foundation models, and enhance them by adding the ability to reliably explore new environments.
>
> **Issue 3: Scalability and Computational Expense**
>
> While the reviewer is right that querying an LLM could be computationally expensive, our algorithm is remarkably fast compared to traditional RL since (1) there are no gradient updates and (2) traditionally RL is notoriously sample-inefficient. Indeed, we exceed the performance of the classic RL GLAM algorithm in Section 4.2 on the order of minutes, whereas the experiments in their paper took >10K GPU hours. Thus, we are actually significantly cheaper in computational cost. We have highlighted this point in Section 5 of the revised paper.
>
> Looking to the future, we have already seen vast improvements to both the capabilities of LLMs and inference speed over the recent years (and corresponding dramatic reductions in cost), and we expect this trend to continue over time. Our new experiments (in Table 2 of the general response) show that Llama3-400B works well with IGE and is more than 10x cheaper. These developments directly improve both the performance and speed of Intelligent Go-Explore, and led to a corresponding **approximately 10x decrease in the cost of IGE**. Moreover, as foundation models (and compute in general) continue to exponentially decrease in cost, IGE will be able to be used in a broader range of important domains.
>
> **Issue 5: Importance of Action Selection**
>
> The reviewer is right that action selection is often the most significant factor in IGE's performance; however, we find that state selection in BabyAI where environment horizons are long also plays a critical role. Equally, without intelligent archive filtering, we would not expect to be able to tackle much larger and more ambitious environments as we would expect the state archive to be overwhelmed in these settings. In contrast, Table 3 shows that we can cut down the state archive size up to 5x in these harder environments, hinting at IGE's scalability to much larger state spaces. We have added this discussion to Appendix C of the revised paper.
>
> **Issue 6: Providing a More Grounded Discussion of Limitations**
>
> We appreciate the feedback and have revised the manuscript to tone down our discussion of these exciting future directions, making it clear that they represent potential avenues for future work that require thorough exploration. We believe many potential extensions of IGE are readily realizable; however, to provide just one example, as suggested by Reviewer xqm5, it could be quite powerful to do imitation learning by fine-tuning models like GPT-4 to distill IGE’s trajectories back into the base foundation model to improve performance. That would be relatively cheap on OpenAI's fine-tuning API. We have revised Section 3 and the conclusion to provide a more grounded discussion of limitations and future work.
>
> **Question 2: Alternate Action Selection Strategies**
>
> We completely agree that random action selection and FM action selection are two extremes. Our framework is completely flexible and we could even consider substituting the action selection policy in IGE with any RL policy. This would enable us to keep the foundation model IGE scaffolding that prevents detachment and derailment and allow us to use much cheaper options to roll out trajectories. We believe this would represent exciting future work and we have augmented the paper with this discussion in Appendix F.
>
> ---
>
> We appreciate your detailed and constructive feedback. Please let us know if we have adequately addressed your concerns? If so, we would appreciate you considering increasing your score. If not, please let us know if there is anything else we can answer and/or address in the remaining discussion period!

---

> ### Comment · Reviewer_ZTB4 · 2024-11-26
>
> Thank you for the rebuttal. However, my concerns remain insufficiently addressed:
>
> - The new "domain-general" prompt is not provided, making it difficult to verify the claim that IGE is independent of prompt engineering. Moreover, crafting factual environment descriptions can still be as, or in certain cases more, effortful as defining heuristics in the original Go-Explore.
>
> - The justification for avoiding Atari benchmarks, where Go-Explore excels, does not address whether IGE genuinely improves upon Go-Explore in true hard-exploration tasks. This remains critical for substantiating the paper's main claim.
>
> - The foundation model-guided action selection is the main driver of IGE's performance, yet it is primarily compared to random actions. Additionally, examples of foundation models excelling in tasks like Doom or robotics, while promising, are external to the paper and do not directly demonstrate whether IGE, as implemented, can be applied to hard-exploration tasks. It feels too speculative to rely on improvements in foundation models or results from other works. Simply observing that GPT-4 outperforms GPT-3 does not imply a steady improvement trend, especially as recent advancements in LLMs have not yielded similarly substantial gains.
>
> While the core ideas of the paper are potentially interesting and suitable for a venue like ICLR, the presentation and claims would benefit from a more modest tone and better evaluations to substantiate its strong claims. For these reasons I willl not raise my score.

---

> ### Author Response · Authors · 2024-11-27
> **Further Reponse to Reviewer ZTB4 (Part 1/2)**
>
> Dear Reviewer ZTB4,
> Thank you for your continued engagement and valuable feedback. We have also uploaded a revised version of the PDF where we have adopted all of your previous suggestions. Below, we address each of your new points.
>
> > 1. The new "domain-general" prompt is not provided, making it difficult to verify the claim that IGE is independent of prompt engineering. Moreover, crafting factual environment descriptions can still be as, or in certain cases more, effortful as defining heuristics in the original Go-Explore.
>
> We apologize for not including the exact "domain-general" prompts earlier. We have now added these prompts to **Appendix C.2** in the revised paper. These prompts are minimal, containing only basic environment descriptions without any task-specific hints or strategies.
>
> Crafting factual environment descriptions requires significantly less effort and domain expertise than designing effective heuristics for Go-Explore. Often, these descriptions can be copied from existing sources like Wikipedia or code repositories. In our case, these are short and took on the order of minutes to copy. For example, the Game of 24 environment description was largely copied from the Yao et al., 2023 (as we explain in the appendix) which took seconds. On the other hand, finding effective heuristics for classic Go-Explore takes many rounds of experiments and iterations as you may not find optimal choices for them straight away. This takes much more time than simply copying and pasting a domain description.
> As we discussed, we believe the success of IGE is independent of what description we choose for the environment. In the experiments in Appendix C.2, we use the same reduced description for all baselines, and IGE strictly improved upon that baseline performance.
>
> > 2. The justification for avoiding Atari benchmarks, where Go-Explore excels, does not address whether IGE genuinely improves upon Go-Explore in true hard-exploration tasks. This remains critical for substantiating the paper's main claim.
>
> We believe the environments we evaluated *are* indeed true hard-exploration tasks. For example, in **BabyAI** and **TextWorld**, state-of-the-art RL (GLAM, ICML 2023)  algorithms that have been published in major ML conference fail to find optimal solutions even after millions of environment steps, and sometimes only on a single seed of the environment. In contrast, IGE solves these tasks efficiently with less than a thousand environment steps, showcasing significant performance gains. These results demonstrate that IGE genuinely improves upon the prior SOTA in challenging exploration settings. Do you disagree that these environments are hard for RL algorithms?
>
> While extending IGE to Atari games is valuable, it would involve significant computational resources (primarily because the environment horizon could be orders of magnitude longer than in the examples that we choose) and is beyond the scope of this paper. It also prices out small academic labs like ours to require these experiments which could take many thousands of dollars. However, we agree with the reviewer that this is an important direction for future research and the acceptance of this paper could allow us to write grants to try to acquire the funds to conduct this extremely important and valuable research.
>
> > 3. The foundation model-guided action selection is the main driver of IGE's performance, yet it is primarily compared to random actions….**
>
> Thank you for this question. We certainly agree that foundation model action selection is at the core of IGE’s performance but we do not believe this is not a downside. Indeed, the literature has shown that designing effective foundation model agents for complex environments is highly non-trivial and we strongly beat SOTA FM agents like ReAct (Yao et al., ICLR, 2023) which uses thought-action prompting and self-reflection agents (Shinn et al., NeurIPS, 2023). Intelligent Go-Explore demonstrates large improvements over these by considering additional mechanisms such as intelligent state selection and archiving in a Go-Explore framework—this helps unlock the remainder of the performance required to consistently get near optimal performance in many environments.
>
> Is there a clear alternative that we should test instead that would work for the environments we choose? In this work, we are trying to as much as possible avoid training any neural network and solve challenging exploration tasks in-context with foundation models in a matter of minutes rather than in hours+ with RL. So is there a control with foundation models you would find more convincing? This is the best and fairest control we could think of. If there is not a better control you can suggest, is it fair to reject our paper on this front?

---

> > ### Author Response · Authors · 2024-11-27
> > **Further Reponse to Reviewer ZTB4 (Part 2/2)**
> >
> > > The presentation and claims would benefit from a more modest tone and better evaluations to substantiate its strong claims
> >
> > We have carefully reviewed the manuscript to adopt a more measured tone in all places you previously identified. Can you please let us know where in the revised  PDF there is language you disagree with, and what you think it should be changed to?
> > We sincerely hope that these revisions address your concerns. We appreciate your time and thoughtful consideration and would be grateful if you could review the revised paper and consider re-evaluating your assessment.
> >
> > Thank you again for your valuable feedback.
> >
> > The Authors

---

> > > ### Comment · Reviewer_ZTB4 · 2024-11-27
> > >
> > > I appreciate the authors adding the domain-general prompts to the paper, which helps address concerns about prompt engineering. However, it is worth noting that these omissions seem to result in quite significant performance drops, challenging the claim of 'IGE being independent of any prompt engineering'.
> > >
> > > To clarify, I do not dismiss the validity or hardness of the environments used in the paper when comparing IGE to other agents. These environments indeed pose challenges and are suitable for testing foundation model-based agents. However, my concern pertains specifically to the comparison made in Table 2, where a direct comparison is made between IGE and the original Go-Explore algorithm in these environments. If the claim is that IGE improves upon Go-Explore, I think it would be important to **also** evaluate this in environments where Go-Explore has been demonstrated to excel, such as the Atari suite.
> > >
> > > I understand that computational resources may limit such an evaluation and I would not expect evaluation across the entire suite, but you could have picked a simple environment like Freeway to make at least one such comparison (or perhaps use MinAtar for further simlpification).
> > >
> > > Since the initial version of the paper had a tendency of making strong claims and overselling its contributions (as other reviewers had also pointed out), but the revised version seems to take on a more modest tone and its contributions are sufficiently interesting for a venue like ICLR, I'm happy to raise my score to 6 - I would not be opposed to acceptance of this version of the paper.

---

> ### Author Response · Authors · 2024-11-27
>
> Dear Reviewer ZTB4,
>
> Thank you so much for your feedback which has helped us improve the paper and for raising your score!
>
> Thank you for your additional comments! Regarding the independence of IGE, we will clarify this further to specifically mean that reductions in prompts effect FM baselines across the board (including Naive LLM, ReAct, Reflexion) but IGE still consistently (statistically significantly) beats them, independent of initial performance.
>
> We are happy that you agree that the current presentation takes a much more measured tone and frames the contributions more accurately. We will also take the suggestions to evaluate on Atari (e.g. Freeway or MinAtar) on board and endeavor to follow up this work with these exciting experiments!
>
> Thanks again!
>
> Best,
>
> The authors

---

### Official Review · Reviewer_xqm5 · 2024-11-02

**Soundness:** 2
**Presentation:** 3
**Contribution:** 3
**Rating:** 8
**Confidence:** 4

**Summary:**

Summary: The authors propose a method that reconceptualises the original Go-Explore method to use the pre-trained knowledge contained in large foundation models.
For this, the original components of Go-Explore such as keeping a state archive and exploring from a previously attained state are defined for a language foundation model.
The idea behind this is that foundation models are better able to reason about what defines an interesting (novel) state or behaviour, while exploiting the creative abilities of foundation models. The authors test on 3 different environments. 2 text based environments and one environment where the visual
Features are described in terms of text.  The authors show that intelligent go-explore is able to solve these environments in very few steps while also outperforming other foundation model based methods.


Overall, this is a well written and interesting paper. I think the idea of creating a foundation model based exploration algorithm that follows along the lines of Go-Explore is clever and perhaps shows a more generalised way to exploit foundation models for exploration.
While the authors ablate their method well, I feel more interesting ablations would concern the capacity of the foundation model itself in terms of how complex the observation space can get and how different prompts, i.e., conditioning affects the exploration. For instance,
Did the foundation model hallucinate during some runs? What do these hallucinations look like? I think this should even be achievable with the presented BabyAI environment. See more detailed info below.

**Strengths:**

- Exploiting the pre-trained biases of LLM for RL is an interesting idea and so far largely unstudied for exploration in reinforcement learning
- I like that the foundation model is defined in terms of a well known exploration algorithm such as Go-Explore, since it shows how concepts from exploration can be transferred to large foundation models.
- The paper is very clear and well written, while providing thorough ablations methods components such as in Section 4.3.

**Weaknesses:**

- This paper argues against hand-crafted features i.e. heuristics, however having to transfer the environments to a prompt based structure is also a hand crafted feature space.
- The paper states in the abstract that the appeal of using foundation models is making use of the foundation models capabilities of reasoning ahead or capitalising on serendipitous discoveries.  While the results are good, can you show examples of trajectories IGE creates compared to a normal RL algorithm that you could argue are serendipitous?

**Questions:**

- How would you transfer this method to continuous state-action spaces? Did you observe a limit regarding how large the state archive can become? For instance int the BabyAI environments the observations are discrete, that is each observation can be mapped to a distinct vector. I imagine this becomes more problematic the larger the environment becomes.
- Did the foundation model hallucinate while creating the trajectories? This would be interesting to see, since failure modes for foundation models seem to differ from traditional RL models.

---

> ### Author Response · Authors · 2024-11-22
> **Response to xqm5 (Part 1/2)**
>
> We thank you for your positive feedback and thoughtful questions. We are pleased that you find "exploiting the pre-trained biases of LLM for RL is an interesting idea and so far largely unstudied for exploration in reinforcement learning," and that "the paper is very clear and well written," providing "thorough ablations" of our method.
>
> **Issue 1: Addressing Hand-Crafted Features in Prompts**
>
> We acknowledge that providing environment descriptions in prompts involves some specification. However, we conducted experiments to assess the robustness of IGE to prompt variations.  Specifically, we removed any hints on how to solve the tasks and kept the prompts purely factually descriptive about the environment. For example, as suggested by Reviewer ZTB4, in the BabyAI environment, we removed statements like "do not repeat actions if they have no effect". The results, shown in Table 1 below, indicate that even with minimal prompts, IGE **strongly outperforms baselines (statistically significant, p < 0.05)** with a domain-general prompt (save for a description of that environment). The evaluation setting is identical to that of Table 2 of our paper. This demonstrates that IGE's effectiveness does not rely heavily on prompt engineering, and the need for hand-crafted prompts is minimal compared to designing heuristics for traditional exploration methods. We have updated the final paper to reflect this finding in Appendix C.2.
>
>    **Table 1: IGE Performance with Domain-General Prompts**
>
>    *Game of 24*
>
> | Model                  | Naive LLM         | ReAct            | Reflexion        | IGE               |
> |------------------------|-------------------|------------------|------------------|-------------------|
> | Original (GPT-4)       | 0.70 ± 0.14       | 0.82 ± 0.12      | 0.83 ± 0.11      | 1.00 ± 0.00       |
> | Domain-General Prompt (GPT-4) | 0.35 ± 0.13 | 0.71 ± 0.13      | 0.71 ± 0.13      | 0.96 ± 0.04       |
>
>    *BabyAI "Put Next To"*
>
> | Model                  | Naive LLM         | ReAct            | IGE               |
> |------------------------|-------------------|------------------|-------------------|
> | Original (GPT-4)       | 0.12 ± 0.12       | 0.48 ± 0.12      | 0.84 ± 0.08       |
> | Domain-General Prompt (GPT-4) | 0.08 ± 0.12       | 0.24 ± 0.12      | 0.68 ± 0.12       |
>
>    *TextWorld "Cooking Game"*
>
> | Model                  | Naive LLM         | ReAct            | Reflexion        | IGE               |
> |------------------------|-------------------|------------------|------------------|-------------------|
> | Original (GPT-4)       | 0.40 ± 0.16       | 0.56 ± 0.16      | 0.52 ± 0.16      | 0.92 ± 0.08       |
> | Domain-General Prompt (GPT-4) | 0.44 ± 0.16       | 0.48 ± 0.16      | 0.48 ± 0.16      | 0.72 ± 0.16       |
>
>
> **Issue 2: Examples of Serendipitous Trajectories**
>
> Thank you for this interesting question! One piece of evidence for seeing the benefit of serendipity relative to standard RL algorithms is the sheer difference in sample efficiency and performance. For example, our algorithm finds solutions to challenging RL environments orders of magnitude quicker than prior RL algorithms (e.g. on BabyAI, GLAM (Carta et al., 2023) fails to find optimal solutions at all (1-4% success rate on the hardest environments) even after millions of environment timesteps). Similarly, in the TextWorld "Coin Collector" domain, previous work (Xu et al., 2020) shows that millions of timesteps are required for agents to find coins in a single seed. In contrast, IGE succeeds in less than a thousand environment steps. This remarkable sample efficiency suggests that our algorithm effectively recognizes key states as they appear and captures serendipity as it happens.
>
> Aside from performance indicating that it must be happening, one could also consider comparing the states selected by hand-designed heuristics (classic Go-Explore) to those discovered by Intelligent Go-Explore and then manually evaluate many to find a few that are surprising. This would be an extremely interesting study on its own but is non-trivial. For example, it requires some level of subjective judgment in choosing which manual heuristics for classic Go-Explore to try. It also requires one to manually review the states found by IGE vs. Go-Explore (and possibly vice versa) and then determine to what degree they count as surprising, to describe why, etc. The flavor of this would be similar to human observers recognizing “Move 37” in AlphaGo was completely unexpected even to Go experts. For all these reasons, we believe that such a study is beyond the scope of this paper and should not be a prerequisite for publication. However, we will mention in our paper that these interesting future research directions are motivated by our work and would be great future research directions.

---

> ### Author Response · Authors · 2024-11-22
> **Response to xqm5 (Part 2/2)**
>
> **Question 1: Transferring IGE to Continuous State-Action Spaces**
>
> Transferring IGE to continuous state-action spaces would be an extremely promising direction! Indeed, we have already observed multimodal foundation models becoming an essential tool in extremely difficult real-world robotics tasks, including “any-to-any” models such as RT-2 [Brohan et al., 2023] and RFM-1 [Covariant AI, 2024] that can reason over “text, images, videos, robot actions, and numerical sensor readings”.
>
> Since the papers cited have already tokenized the state and action spaces of general environments into an FM and shown the ability of the model to generate actions for any state, it should be simple to then ask the FM to judge the interestingness of any given state compared to prior states observed, which is the only additional requirement of IGE. We have added this discussion to the conclusion of the revised paper.
>
> **Question 2: Managing the Size of the State Archive**
>
> IGE intelligently manages the size of the state archive in the TextWorld domain by using the foundation model to filter and retain only the most interesting and promising states, preventing the archive from becoming unmanageable (Section 3.3). Our experiments show that intelligent archive filtering significantly reduces the number of states while maintaining performance (see Table 3 in the paper and Appendix C). We did not observe significant issues related to archive size in our experiments. We have made the existing discussion in Section 5 more clear and added details in Appendix C.
>
> **Question 3: Handling Hallucinations During Trajectory Creation**
>
> The reviewer is right that foundation models have been known to hallucinate. However, in IGE, the foundation model does not create any trajectory (but is rather queried for state/action choices from real options) and exactly saves found states into an archive like the original Go-Explore. Therefore, IGE provides robust scaffolding for foundation model agents that explicitly prevent the hallucination of impossible states.
>
> There is a concern that the LLM could pick impossible actions, but we observe that malformed inputs only consist of less than 0.1% of total actions. We discuss this in Appendix C, in these rare cases, we can simply take the simple choice of randomly sampling an action from the environment.
>
> ---
>
> We appreciate your constructive feedback, which helped us improve the paper by clarifying how IGE handles prompt variations and potential hallucinations. These contributions enhance the understanding of IGE's advantages and robustness. Please let us know if we have adequately addressed your concerns? If so, we would appreciate you considering increasing your score. If not, please let us know if there is anything else we can answer and/or address in the remaining discussion period!

---

> > ### Comment · Reviewer_xqm5 · 2024-11-25
> >
> > I appreciate the clarifications and further test results. I am happy to raise my score, since I think this is an overall good paper.
> > Again, I like that this paper proposes a structured approach to exploration with foundation models based on Go-Explore, which is a general framework that has worked well in the past. It is interesting to see that these mechanisms seem to work in a different feature space with success.
> >
> > I would suggest though, to remove the comment about "serendipitous trajectories" and replace it with your more detailed response. I understand what you mean, but the reasons for the increased robustness of the proposed method should be based on more substantiated observations, as the ones you have written in your response rather than an imprecise statement.

---

> > > ### Author Response · Authors · 2024-11-25
> > > **Thank you!**
> > >
> > > Dear Reviewer xqm5,
> > >
> > > Thank you so much for your feedback which has helped us improve the paper and for raising your score to 8!
> > >
> > > We agree with the suggestion, and will add this more detailed response to the paper. Together with the feedback from Reviewer iSQ8, we will make the language around serendipity far more clear and precise.
> > >
> > > Best,
> > > The authors

---

### Official Review · Reviewer_cN7x · 2024-11-04

**Soundness:** 3
**Presentation:** 3
**Contribution:** 3
**Rating:** 8
**Confidence:** 3

**Summary:**

The paper revisits the Go-Explore (GE) algorithm (Ecoffet et al., 2021) by letting a foundation model (FM) control three key operations performed on an archive of interestingly new states: (a) select a state to explore from (b) choose actions from the selected state, and (c) decide which state is worth keeping. The paper argues that FM successfully replaces GE's requirement to use hand-designed heuristics domain-specific pre-specified knowledge. IGE, with GPT4 as the underlying LLM, is evaluated on three families of environments (Game of 24, BabyAI, and TextWorld). It is claimed that the method outperforms baselines consistently across the board, sometimes being the only method that shows life (e.g., for Coin Collector). Finally, the following ablation studies are provided: (a) the importance of FM, (b) the size of the archive, and (c) the capabilities of FM.

**Strengths:**

* The paper considers a fundamental problem of managing the exploration-exploitation trade-off.
* IGE proposes to leverage the capabilities of LLMs to address deficiencies in the original GE and shows that it succeeds in three families of environments.
* The paper fits into the important line of work concerned with applications of foundation models in RL.

**Weaknesses:**

* The choice of a setup for IGE matters, as it seems to work for deterministic environments. Is the extension to stochastic environments a fundamental limitation of IGE, or is there a "natural" extension? One example could be NetHack, which is a known hard exploration problem with long-horizon and credit assignment challenges.
* The IGE's procedure of selecting interesting states from the archive resembles a hierarchical search, where the planning is performed on high-level milestones or subgoals. Often such methods use Best-First-Search as a backbone planner, hence they jump freely between subtrees, similarly as IGE (see a discussion in lines 298-302 or 490-496). Can the Authors comment on that?
* Related to the above, how does IGE compare against algorithms like MCTS for hard deterministic combinatorial problems (like chess or go)?
* What is the relation of IGE to GFlowNets, which searches through a design space to generate samples proportional to a specified reward function?
* What important real-life problems can be expressed as exploration and solved via IGE? Say, interesting chess or go states, novel mathematical theories or proof ideas, useful molecules, etc.
* Can the method accelerate research on open-endedness, which is often formulated as an unsupervised exploration problem?
* How robust is the method?
	* LLMs are known to be unreliable and have a long list of failure modes.
	* Are there any biases with respect to some environments games, i.e., the method performing better in the environments that were prominently represented in the pre-training dataset?
	* Conversely, does IGE underperform for environments that were underrepresented in the training dataset or that require a set of rules or wiki?
* How can IGE be augmented with the inclusion of environment rules, RAGs, and tools? How hard is it to set up IGE for a new task? How much work is expected on the prompt engineering part?
* What is the performance of open-sourced models?
* IGE is an in-context method. What could be achieved that currently is out of reach if the method allowed fine-tuning (or RL training)?
* What could be the implications for the current RL algorithms regarding the potential benefits stemming from IGE providing better data?
* Can we formulate an inverse-IGE problem by designing synthetic datasets to discover the "definition of interestingness of LLMs"?


Other:
* In Algorithm 1, lines 3, 8, 10, should account for the fact that "$\gets$" typically stands for assignment operator, not "add to a set" operator.

Edit 27th November 2024: Based on the Authors' response, I have increased the score to 8.

**Questions:**

See above

---

> ### Author Response · Authors · 2024-11-22
> **Response to cN7x (Part 1/3)**
>
> We thank the reviewer for their extensive feedback and insightful questions. We appreciate that you recognize our paper "considers a fundamental problem of managing the exploration-exploitation trade-off" and "fits into the important line of work concerned with applications of foundation models in RL".
>
> **Question 1: Extension to Stochastic Environments**
>
> We appreciate this important question. We believe that IGE can be extended to stochastic environments, similar to how the original Go-Explore (Ecoffet et al., 2019) was extended (Ecoffet et al., 2021). In the 2021 Go-Explore work, a robust goal-conditioned policy is trained via imitation learning on the trajectories found during exploration, enabling generalization to stochasticity. In the context of IGE, we can similarly collect successful trajectories and provide them in-context to the foundation model, leveraging its ability to generalize and handle stochastic outcomes. We have added a discussion on this potential extension in Appendix F of the revised paper.
>
> **Question 2: IGE Resembling Hierarchical Search and Best-First Search**
>
> Indeed, the Go-Explore framework, and by extension IGE, shares similarities with hierarchical search and Best-First Search, as it prioritizes exploration from the most promising states. However, IGE builds on this approach by leveraging the foundation model's intelligence to dynamically assess the interestingness and potential of states, rather than relying on fixed heuristics. This allows IGE to adaptively explore the search space in a more informed manner. We have added a comparison and discussion of this relationship in Appendix F of the revised paper.
>
> **Question 3: Comparison to MCTS**
>
> Thank you for the fascinating question! For these extremely hard combinatorial problems like chess and Go, you would likely need an extremely good interestingness function. This relates to your later question on whether we can extract good interestingness functions from foundation models. If we could train one or otherwise, combined together with a solid value function just like in MCTS, IGE could work extremely well and discover novel and interestingly new strategies. We have added this discussion to Appendix F of the revised paper.
>
> **Question 4: Relation to GFlowNets**
>
> GFlowNets aim to sample compositional structures proportionally to a specified reward function, which differs from IGE's goal of exploring and discovering interesting states without predefined rewards. While both involve generating states through sequential decisions, IGE focuses on leveraging the foundation model's notions of interestingness rather than sampling according to a reward distribution. Moreover, GFlowNets are not directly intended for hard-exploration/sparse-reward problems. IGE should therefore be much better in hard exploration tests where there is little to no reward signal to guide search. We have added a comparison to GFlowNets in Appendix F of the revised paper.
>
> **Question 5: Real-life Applications of IGE**
>
> IGE's framework is very general, and we envision that it could be extended to many real-life problems involving exploration in complex spaces, such as discovering novel proteins in synthetic biology, or exploring mathematical conjectures. Recent work has already begun to explore applications of LLM agents across science, for example, Laurent et al. (2024) for biology research and AlphaGeometry (2024) for olympiad-level math problems.
>
> On the practical side, LLMs have been adapted for various web-browsing and computer-based tasks (Liu et al., 2023) as useful personal assistants, many of which require exploration across long horizons (e.g., building an app from scratch).
>
> We have expanded the existing discussion on this in the conclusion and Appendix F to add these exciting points!
>
> **Question 6: Accelerating Open-Endedness Research**
>
> We agree! The reviewer is right that open-endedness is often formulated as an unsupervised exploration algorithm, so we should aim to develop the best and most general open-ended exploration algorithms possible. Our algorithm provides a general mechanism to explore and discover a diverse set of interesting states or solutions in arbitrary environments without predefined objectives, much like human scientific exploration. We have added a discussion on this potential in Appendix F of the revised paper.

---

> ### Author Response · Authors · 2024-11-22
> **Response to cN7x (Part 2/3)**
>
> **Question 7: Robustness and Potential Biases**
>
> This is a great question. Although representation in pre-training is hard to measure even for open-weight models, we have strong evidence that IGE’s success is relatively independent of the base LLM performance. For instance, in our new results in Table 2 of the general response, with Claude Sonnet 3.5 on Game of 24, the naive action selection performance is low (0.24), but IGE raises it significantly to 0.86.
>
> The general question of bias is a very interesting one; for example, if a foundation model was biased to think that nothing “purple” is interesting, then we wouldn’t explore states that are “purple”. We do not observe such issues in our current evaluation (indeed, we would expect them to manifest less in games and RL/control problems rather than human-centric problems), but this raises an interesting point that is tied to the wider literature of reducing bias in foundation models in general. We have expanded on this in Appendix F of the revised paper.
>
> **Question 8: Ease of Setting Up IGE and Incorporating Additional Tools**
>
> Setting up IGE for a new task involves providing a brief environment description (e.g., from Wikipedia or a codebase which should require minimal knowledge of the domain). Importantly, our experiments with domain-general prompts (see Table 1 in the general response) demonstrate that IGE's success is independent of any domain-specific prompt engineering. We removed any hints about how to solve the tasks and kept the prompts purely factually descriptive. Even with minimal prompts, IGE substantially outperforms baselines, indicating that per-domain prompt engineering is not necessary.
>
> Furthermore, since we use generic foundation model APIs to select novel states, actions, or when to archive, we can readily integrate further tools into the LLM calls, which could allow us to use things like web search for more hints or tools to do complex processing with ease. As we discuss in the paper, IGE is an extremely generic framework for exploration that is strictly additive on top of any foundation model strategy. We have added this discussion to Section 3 of the revised paper.
>
> **Question 9: Performance with Open-Source Models**
>
> We evaluated IGE using other foundation models, including *Claude Sonnet 3.5* and the **open-weight** *Llama-3 400B*, to address concerns about IGE's dependence on specific models. The results, presented in Table 2 of the general response, show that IGE consistently outperforms baselines **(statistically significant, for all baselines with Llama-3 400B and for all except Reflexion on Sonnet 3.5, *p* < 0.05)** across different models, demonstrating that IGE is a generic improvement that can be used in conjunction with any foundation model, likely including as new models are released that are even more powerful. The evaluation setting is identical to that for Table 2 of our paper. We have added all of these results to Section 5 of the revised paper.
>
>    **Table 2: IGE Performance with Different Foundation Models on Game of 24**
>
> | Model                | Naive LLM         | ReAct            | Reflexion        | IGE               |
> |----------------------|-------------------|------------------|------------------|-------------------|
> | GPT-4                | 0.70 ± 0.12       | 0.82 ± 0.11      | 0.83 ± 0.10      | 1.00 ± 0.00       |
> | Claude Sonnet 3.5    | 0.24 ± 0.12       | 0.58 ± 0.14      | 0.80 ± 0.11      | 0.86 ± 0.09       |
> | Llama-3 400B         | 0.44 ± 0.14       | 0.68 ± 0.13      | 0.54 ± 0.14      | 0.98 ± 0.03       |
>
> **Questions 10 and 11: Potential of Fine-Tuning or RL Training**
>
> Fine-tuning or additional RL training could potentially enhance foundation models' performance in specific tasks. In the context of IGE, one could envision fine-tuning the foundation model on the trajectories discovered during exploration, further improving its decision-making and ability to generalize. However, our current approach shows that significant gains can already be achieved without additional training, which is advantageous for efficiency. We discuss this in Section 3: "These solutions could easily then subsequently be used for downstream reinforcement learning or even improve the foundation model in the next task by in-context learning—thus allowing an agent to bootstrap its own learning indefinitely."
>
> Furthermore, there could be equivalent benefits for even non-LLM-based RL algorithms. The trajectories that are discovered by IGE could be fed into a traditional imitation learning algorithm just as in the original Go-Explore. Or we could do offline RL on diverse data collected all throughout exploration and hope to get even better policies. We have added this discussion to Appendix F of the revised paper.

---

> ### Author Response · Authors · 2024-11-22
> **Response to cN7x (Part 3/3)**
>
> **Question 12: Investigating LLMs' Notions of Interestingness**
>
> Understanding and formalizing the foundation model's notions of interestingness is indeed an intriguing direction for future research. In a similar way that reward models are collected on human preferences and used to fine-tune LLMs, we could also consider extracting interestingness preferences and fine-tuning an LLM to select more interesting states or actions. This could then lead to far more efficient versions of IGE, or even improvements to FMs themselves. We have added this discussion to the future work section in Appendix F of the revised paper.
>
> **Minor Comment on Notation**
>
> Thank you for suggesting to use the assignment operator in Algorithm 1. We have corrected the notation to accurately represent the addition of elements to a set in Algorithm 1 of the revised paper.
>
> ---
>
> We appreciate your comprehensive and thoughtful questions, which have helped us expand and clarify important aspects of our work. Please let us know if we have adequately addressed your concerns? If so, we would appreciate you considering increasing your score. If not, please let us know if there is anything else we can answer and/or address in the remaining discussion period!

---

> > ### Author Response · Authors · 2024-11-27
> >
> > Dear Reviewer cN7x,
> >
> > Thank you for your insightful feedback and for engaging in the discussion. We wanted to inform you that we have updated the PDF of our paper to incorporate all your suggestions and address your concerns thoroughly. Your comments have significantly improved the quality and clarity of our manuscript, and inspired future work that we have discussed at length.
> >
> > We would greatly appreciate it if you could review the revised paper and let us know if we have adequately addressed your feedback. If so, we would be grateful if you would consider raising your score, especially since we guess that the paper’s current scores make it “on the fence” in terms of whether it will be accepted for publication. If there are any remaining issues or additional suggestions, we welcome your further input. Your perspective is invaluable to us, and we aim to make the paper as strong as possible.
> >
> > Thank you again for your time and thoughtful review.
> >
> > Sincerely,
> >
> > The Authors

---

> > > ### Comment · Reviewer_cN7x · 2024-11-27
> > >
> > > I thank the Authors for their thorough response and added discussion to the revised version of the paper. I decided to increase the score 6->8.

---

> > > > ### Author Response · Authors · 2024-11-27
> > > >
> > > > Dear Reviewer cN7x,
> > > >
> > > > Thank you so much for your feedback which has helped us improve the paper's clarity and experimental rigor and for raising your score to 8!
> > > >
> > > > Best,
> > > >
> > > > The authors

---

### Official Review · Reviewer_iSQ8 · 2024-11-04

**Soundness:** 2
**Presentation:** 2
**Contribution:** 3
**Rating:** 6
**Confidence:** 3

**Summary:**

Combines the Go-Explore -- a framework for solving hard-exploration problems -- with language models (LMs) to create LM agents that do better on search and exploration tasks.

**Strengths:**

**[Originality]**
The idea seems original, and creatively combines Go-Explore (a traditional RL approach for exploration) and LM agent tasks.

**[Quality]**
The method shows strong performance over its baselines.


**[Clarity]**
The prompting procedures are well described and environments are well described

**[Significance]**

The combination of RL exploration methods and LM agents feels both natural, thus this work opens up the possibility to further explore and improve components of the Go-Explore framework for better LM-agents.

**Weaknesses:**

While I quite enjoyed the idea and appreciate that using LMs with the go-explore framework is a promising direction, the main weaknesses with the current version of the paper is that many of the stated / emphasized claims are quite strong, which I do not feel they are properly back-ed up. See below.

> IGE ofers the previously impossible opportunity to recognize and capitalize on “serendipitous discoveries” (L24, L76, L95)

- The use of the word “serendipitous” feels un-rigorous and only added confusion. I don't know what this word means in context, nor how it contributes scientifically to this paper.
- Based on my reading, the authors generally use this word when describing the ability to identify which states should be archived to later explore further. This is an improvement over the original Go-Explore which use a heuristic (cell representation) instead of LM-preference. Perhaps it is better to just state this as-is.
- Further, the claim that this was “previously impossible” feels very strong. It is also unclear *what* was previous impossible? There exist previous work that use LM as a feedback to derive intrinsic motivation (e.g. Klissarov et al 2023, which the authors cite, L474).

> IGE integrates well with various agent strategies ... and will only get better as capabilities of foundation models improve further (L103)

- The experimental evidence for this claim is in Table 3 right, which shows IGE does better on GPT-4 than GPT-3.5. One hypothesis is that IGE gets better as foundation models improve; another hypothesis could be that IGE is only well-tuned to GPT-4 and may not work in other / later models. Experiments across multiple foundational models with different capabilities will make this more convincing.

> Key strength of IGE is that it is a strict improvement on top of any FM agent framework (L195)

- Unclear if the authors mean IGE is *complementary* to other FM agent frameworks; or that IGE will always be the best FM agent framework?


The strong claims contrast against the empirical results, which are good, but marginally so (e.g. between Fig 3 and 4, IGE is statistically significantly better than other baselines in 3 out of 13 settings between Figures 3 and 4 (evaluated base on overlapping 95 confidence interval bars). I feel that a more measured presentation can aid in paper clarity.

**Questions:**

1. Can the authors provide more details of how baselines are run? For instance, do all LM baselines get the same environment description for each game?

2. Are there ablations of sensitivity of the approach to differences in prompts?

3. How exactly is the state conditional history (L199) implemented? Does it store a set of previously attempted actions for each seen state? Is it a buffer of (state, action) pairs?

4. Do you think the exploration problems Go-Explore addresses are also problems in the evaluation tasks? Specifically, detachment and derailment (L130)?

5. How is state reset done (i.e. to explore from a certain state) in the provided environments?

6. In table 2, is there a difference between “classic go explore” and “ablated all 3 above” models?

7. Just out of curiosity, there do exist work that tackle exploration very explicitly in difficult games such as MineCraft [Wang 2023, Zhu 2023]. Is there a reason these methods are *not* reasonable comparisons against IGE?


[Wang 2023] Wang, Guanzhi, et al. "Voyager: An open-ended embodied agent with large language models." arXiv preprint arXiv:2305.16291 (2023).

[Zhu 2023] Zhu, Xizhou, et al. "Ghost in the minecraft: Generally capable agents for open-world environments via large language models with text-based knowledge and memory." arXiv preprint arXiv:2305.17144 (2023).

---

> ### Author Response · Authors · 2024-11-22
> **Response to iSQ8 (Part 1/3)**
>
> We thank the reviewer for their insightful feedback and for recognizing the originality and strong performance of our method. Below we address each of your comments. We hope you do not mind if we mention that we were surprised by how low your score is given your review, as it seemed that the issues you raised, while important, were not in line with that low of a score. The other reviewers gave the paper much higher scores (and offered to further increase them if we address their issues, which we have). All told, in light of all of this we hope you will consider increasing your score, as right now it makes the paper unlikely to be published and shared with the ICLR community. We (and the other reviewers) do think it makes a valuable contribution that should be published, and we hope you will also agree after all of the improvements we have made.
>
> **Issue 1: Clarification of "Serendipitous Discoveries"**
>
> We apologize for any confusion caused by the term "serendipitous discoveries". In the context of IGE, by "serendipitous discoveries" we refer to unexpected but valuable states that are encountered during exploration without prior anticipation by the human user. The original Go-Explore relied on hand-crafted heuristics to define ahead of time what types of states would count as interestingly new, which thus had no ability to recognize a state that is surprising, interesting, and potentially useful in an unanticipated way (e.g., a hand-coded heuristic might have declared that new x,y locations in the game and differing numbers of keys are interesting, but thus could not recognize that drinking a potion and then tripling in size constitutes a new, interesting state). IGE leverages the foundation model's internalized human notions of interestingness to recognize and add even entirely unanticipated novel states to the archive. This improves exploration by identifying valuable stepping stones that might otherwise be missed. As we quoted in the introduction, in the words of Isaac Asimov—“The most exciting phrase to hear in science, the one that heralds new discoveries, is not 'Eureka!' but 'That’s funny'." We have clarified this definition in the revised paper, specifically in Section 1 (Introduction).
>
> **Issue 2: Adjusting Strong Claims on "Previously Impossible"**
>
> We agree that the phrasing "previously impossible" was not clear, and too strong without the proper scope. What we meant is that this was previously impossible within the Go-Explore framework. In comparison to Klissarov et al. (2023), our algorithm requires no training and finds solutions to hard RL environments within minutes. We have adjusted the language in the revised paper to more accurately reflect this, in Section 1 (Introduction).
>
> **Issue 3: Claims about Performance Scaling with Foundation Models**
>
> Thank you for this suggestion to include experiments with non-GPT foundation models. We evaluated IGE using other models (**Claude Sonnet 3.5** and **Llama-3 400B**) on the Game of 24. The results, presented in Table 2, show that IGE consistently outperforms baselines **(statistically significant, for all baselines with Llama-3 400B and for all except Reflexion on Sonnet 3.5, p < 0.05)** across different models, demonstrating that IGE is a generic improvement that can be used in conjunction with any foundation model, likely including as new models are released that are even more powerful. We have included these findings in Section 5 of the revised paper.
>
> **Table 2: IGE Performance with Different Foundation Models on Game of 24 (Same Eval Setting as for Table 2 of the original submission)**
>
> | Model                | Naive LLM         | ReAct            | Reflexion        | IGE               |
> |----------------------|-------------------|------------------|------------------|-------------------|
> | GPT-4                | 0.70 ± 0.12       | 0.82 ± 0.11      | 0.83 ± 0.10      | 1.00 ± 0.00       |
> | Claude Sonnet 3.5    | 0.24 ± 0.12       | 0.58 ± 0.14      | 0.80 ± 0.11      | 0.86 ± 0.09       |
> | Llama-3 400B         | 0.44 ± 0.14       | 0.68 ± 0.13      | 0.54 ± 0.14      | 0.98 ± 0.03       |
>
>
> **Issue 4: Clarification on IGE's Complementarity to Other Frameworks**
>
> We agree that "strict improvement on top of any FM agent framework" is unclear. We have changed this in the revised paper to reflect that IGE is complementary to various FM agent reasoning strategies, such as zero-shot, few-shot, or chain-of-thought prompting. IGE integrates these agentic strategies with a structured exploration mechanism grounded in the Go-Explore framework, enabling better performance. We have clarified this in Section 3 of the updated manuscript.

---

> ### Author Response · Authors · 2024-11-22
> **Response to iSQ8 (Part 2/3)**
>
> **Issue 5: Clarifying Empirical Results**
>
> Thank you for raising this point. While the performance gains may appear modest in some easier environments where baseline methods already perform well, IGE shows significant improvements in harder environments where exploration is more challenging.
>
> For instance, our algorithm finds solutions to challenging RL environments orders of magnitude quicker than prior RL algorithms (e.g., on BabyAI, GLAM (Carta et al., 2023) fails to find optimal solutions at all (1-4% success rate on the hardest environments) even after millions of environment timesteps). In contrast, we find solutions on the order of minutes with less than a thousand environment steps. We have revised our presentation in Section 5 to highlight the significance of these results.
>
> Similarly, in the TextWorld "Coin Collector" domain, previous work (Xu et al., 2020) shows that millions of timesteps are required for agents to find coins in a single seed. In contrast, IGE also succeeds in less than a thousand environment steps. This sample efficiency shows a remarkable improvement over prior SOTA in RL.
>
> **Question 1: Details on Baselines and Environment Descriptions**
>
> Yes, all LM baselines receive the same environment descriptions and observations for each game. This ensures a fair comparison, as any differences in performance are due to the methods themselves rather than discrepancies in the information provided. We have clarified this in Section 4 of the revised paper.
>
> **Question 2: Sensitivity to Prompt Variations**
>
> Thank you for this important question. We performed experiments to assess the sensitivity of IGE to prompt variations. Specifically, we removed any hints on how to solve the tasks and kept the prompts purely factually descriptive about the environment. For example, as suggested by Reviewer ZTB4, in the BabyAI environment, we removed statements like "do not repeat actions if they have no effect". The results, shown in Table 1 below, indicate that even with minimal prompts, IGE **strongly outperforms baselines (statistically significant, p < 0.05)** with a domain-general prompt (save for a description of that environment). The evaluation setting is identical to that of Table 2 of our paper. This demonstrates that IGE's effectiveness is robust to prompt variations and does not rely on extensive prompt engineering. We have included these findings and full details in Appendix C.2 of the revised paper.
>
>    **Table 1: IGE Performance with Domain-General Prompts**
>
>    *Game of 24*
>
> | Model                  | Naive LLM         | ReAct            | Reflexion        | IGE               |
> |------------------------|-------------------|------------------|------------------|-------------------|
> | Original (GPT-4)       | 0.70 ± 0.14       | 0.82 ± 0.12      | 0.83 ± 0.11      | 1.00 ± 0.00       |
> | Domain-General Prompt (GPT-4) | 0.35 ± 0.13 | 0.71 ± 0.13      | 0.71 ± 0.13      | 0.96 ± 0.04       |
>
>    *BabyAI "Put Next To"*
>
> | Model                  | Naive LLM         | ReAct            | IGE               |
> |------------------------|-------------------|------------------|-------------------|
> | Original (GPT-4)       | 0.12 ± 0.12       | 0.48 ± 0.12      | 0.84 ± 0.08       |
> | Domain-General Prompt (GPT-4) | 0.08 ± 0.12       | 0.24 ± 0.12      | 0.68 ± 0.12       |
>
>    *TextWorld "Cooking Game"*
>
> | Model                  | Naive LLM         | ReAct            | Reflexion        | IGE               |
> |------------------------|-------------------|------------------|------------------|-------------------|
> | Original (GPT-4)       | 0.40 ± 0.16       | 0.56 ± 0.16      | 0.52 ± 0.16      | 0.92 ± 0.08       |
> | Domain-General Prompt (GPT-4) | 0.44 ± 0.16       | 0.48 ± 0.16      | 0.48 ± 0.16      | 0.72 ± 0.16       |
>
>
> **Question 3: Implementation of State-Conditional Action History**
>
> The state-conditional action history in IGE stores the set of actions previously attempted from each archived state. For each state in the archive, we maintain a list of actions that have been tried from that state, which helps prevent the foundation model from repeating the same actions and encourages exploration of new actions. This history is provided in the context when prompting the foundation model for the next action. This is relatively lightweight since our state archive is kept at a manageable size, as can be seen in Table 3 of our submitted paper. We have added a detailed explanation in Appendix D.3 of the revised paper.

---

> ### Author Response · Authors · 2024-11-22
> **Response to iSQ8 (Part 3/3)**
>
> **Question 4: Exploration Challenges in Evaluation Tasks**
>
> Yes, the exploration challenges that Go-Explore addresses, such as detachment and derailment, are present in our evaluation tasks. For example, in the TextWorld "Coin Collector" environment, the agent navigates a maze-like structure where it can easily prematurely abandon promising initial paths (detachment) or struggle to return to previously visited rooms (derailment). Just as in the original GE, IGE mitigates these issues by archiving interesting states and reliably returning to them for further exploration, guided by the foundation model's intelligence.
>
> **Question 5: State Reset Mechanism**
>
> In our experiments, we assume the ability to reset the environment to any previously visited state, as in the original Go-Explore. For the explicit RL "gym"-style environments like BabyAI and TextWorld, this is straightforward and already implemented. In the text-based Game of 24, we build an environment that can recreate the state based on the archived information. Just as in the original Go-Explore, future extensions of IGE could consider prompting an LLM with past trajectories to return to desired states without the need for explicit reset mechanisms. We have clarified this in Section 2 of the revised paper.
>
> **Question 6: Difference Between "Classic Go-Explore" and "X All 3 above"**
>
> "Classic Go-Explore" refers to the original algorithm using hand-designed heuristics such as selecting states based on visitation counts, taking random actions, and adding unique states to the archive. In contrast, the "X All 3 above" variant in our ablation study replaces the foundation model guidance in IGE with unintelligent choices like random state selection (and the same unintelligent choices for actions and archiving). This helps demonstrate the importance of each component of IGE by comparing it to both the original Go-Explore and a fully ablated version. We have clarified this in Section 5 of the revised paper.
>
> **Question 7: Comparison with Exploration Methods in Games Like Minecraft**
>
> Thank you for highlighting these works. The focus and methodology of these works are quite different. Both Wang et al. (2023) and Zhu et al. (2023) employ agents that act in the environment via collections of high-level code or algorithmic policies tailored to Minecraft. In contrast, IGE provides a generic method to operate in any state and action space, given a minimal description of the environment, by leveraging the intelligence of foundation models at test time. The code policies used in these Minecraft agents are indeed interesting and could potentially be integrated into an IGE-like framework, representing a promising direction for future research into more efficient exploration agents. We have added a discussion on this in Appendix F of the revised paper.
>
> ---
>
> We appreciate your detailed and thoughtful feedback, which helped us improve the paper by clarifying key concepts and addressing your concerns about scaling and methodology. Please let us know if we have adequately addressed your concerns? If so, we would appreciate you considering increasing your score. If not, please let us know if there is anything else we can answer and/or address in the remaining discussion period!

---

> ### Author Response · Authors · 2024-11-27
>
> Dear Reviewer iSQ8,
>
> Thank you for your detailed review and feedback. As we approach the end of the discussion period, we notice that you have not yet responded to our detailed answers to your concerns. Your score differs substantially from other reviewers who rated the paper considerably more favorably (with ratings of 6, 8, and 8). Since ICLR typically requires consensus among reviewers for acceptance, your current score could prevent the paper from being published despite the positive feedback from other reviewers.
> We have uploaded a revised PDF and worked diligently to address each of your concerns by:
>
> - **Clarifying the definition of "serendipitous discoveries":** Added a precise and rigorous explanation in Section 1 to remove any ambiguity.
> - **Adjusting strong claims throughout the paper:** We revised language to avoid overstating our contributions, ensuring all claims are accurate and appropriately scoped.
> - **Adding experiments with different foundation models:** We conducted new experiments using Claude Sonnet 3.5 and Llama-3 400B, demonstrating that IGE consistently outperforms baselines across various models (Section 5).
> - **Conducting experiments on prompt robustness:** We performed experiments with domain-general prompts, showing that IGE's effectiveness does not rely on extensive prompt engineering (Appendix C.2).
> - **Clarifying how IGE is complementary to other frameworks:** We explained in Section 3 that IGE integrates well with various agent strategies and provides general improvements over existing methods.
> - **Enhancing the presentation of empirical results:** We highlighted the significance of our findings in Section 5, especially in challenging environments where baseline methods perform poorly.
> - **Providing detailed responses to your specific questions:** We addressed all your other queries, including baseline implementations, sensitivity to prompts, state-conditional action history, exploration challenges, state reset mechanisms, ablation studies, and comparisons with other methods.
>
> Given these improvements and our detailed responses, would you be willing to review our rebuttal and reconsider your rating? If there are remaining concerns that would need to be addressed to merit a higher score, we would greatly appreciate your feedback before the rebuttal period ends.
>
> Thank you, the authors.

---

> > ### Comment · Reviewer_iSQ8 · 2024-11-28
> >
> > I thank the authors for the response. I appreciate the additional experiments with Claude Sonnet 3.5 and Llama-3 400B, the ablations with less domain specific prompts, the additional context of BabyAI’s PutNextTo and TextWorld’s Coin Collector as environments previous LM-agents struggle with, and in general softening the language of the paper.
> >
> > **Regarding my initial “surprisingly low” score**, my concern was _the paper was not measured or precise enough in its scientific communication_. Unsubstantiated claims such as “offers the previously impossible ability to recognize serendipitous discoveries”, “strict improvement over any FM agent”, along with many speculative claims about IGE’s _potential_ abilities (some examples below), distracted from the actual technical contributions of this work. It gives the reader a feeling that the paper is not an impartial, objective and rigorous investigation of the method, making it unproductive for the community to build on as people cannot accurately evaluate the pros and cons of this method. _That said, I acknowledge that in the new pdf version, the unsubstantiated claims (e.g. “previously impossible”, “strict improvement over any FM agent”) have been fixed._
> >
> >
> > **A few additional suggestions on new pdf draft**
> >
> > - Re: “serendipitous discoveries” (L024), I still think using this term not useful, as it is just too scientifically imprecise. As a lay term with lots of everyday meanings, "serendipity" only distracts the reader from the actual technical contributions of this method. In the paper, “serendipity” is invoked to mean “using LM preference to identify states to cache for future exploration”. The author can just say so, e.g.  “we alleviate the need to rely on human designed heuristics by using LM preference”
> >
> > - “In the words of Isaac Asimove … ‘That’s funny’” (L92) I think Asimov quotes should be removed altogether, though maybe this is more a matter of taste. I personally think this is unnecessary flourish and its location in the text obfuscates the direct comparison between Original GE and IGE.
> >
> > - “Will only get better as the capabilities of FMs continue to improve” (L105) should be softened further. Empirical results in the paper may suggest this, but this is not a fact, it's the author's belief.
> >
> > - “To tackle virtually any type of problem” (L114) should be softened. Only text-based environments are addressed in this work, IGE's applicability to other environments is the author’s belief.
> >
> > - Showing the full trace of how the agent reasons at each step in text (lines 5-10 of Algorithm 1) would be helpful (Figure 4 in [Zhu 2023] does this well) to further clarify the method
> >
> > - L50: should probably just cite GLAM (the only RL method compared against in the paper), not Sutton & Barto. GLAM is _one_ instantiation of an RL + LLM approach and should not be representative of the entire family or RL approaches (this can be mis-leading since the original Go-Explore is compared against many other RL approaches)
> >
> > All in all, **_I think a more measured take-away a reader should walk away with is_**:
> > - **In text-based environments _that allow for reset to to any state_**, IGE can be applied to to _improve the exploration capability of LM agents_
> > - Concretely, this is demonstrated over LM approaches ReAct, Reflexion, and GLAM (albeit GLAM only in the BabyAI-Text environment)
> > - **Whether IGE works well outside of these reset-able text-based environments** (including ones the original Go-Explore excelled in, such as the Atari suite) **remains an unanswered question**. These are **interesting future opportunities** for the community to explore.
> >
> > With the above said, I do think the added results are convincing within the context of the environment tested, and that overall the idea is quite interesting. Thus, I raise my score. Good luck!
> >
> > ---
> >
> > [Zhu 2023] Zhu, Xizhou, et al. "Ghost in the minecraft: Generally capable agents for open-world environments via large language models with text-based knowledge and memory." arXiv preprint arXiv:2305.17144 (2023).

---

> > > ### Author Response · Authors · 2024-11-30
> > >
> > > Dear Reviewer iSQ8,
> > >
> > > Thank you for your continued detailed feedback and for raising your score! We are grateful for your recognition of the additional experiments and revisions we've incorporated into the paper. While we are no longer able to update the PDF because of the ICLR rules, we will revise the paper again in light of your most recent round of feedback. We are confident the paper has been much improved by your feedback, and we deeply appreciate the time you put into reviewing it.
> > > We would like to correct just one point of confusion:
> > >
> > > > "'Only text-based environments are addressed in this work, IGE's applicability to other environments is the author’s belief."
> > >
> > > We would like to gently clarify that, in addition to text-based environments, we have also demonstrated IGE's applicability in a visual environment, specifically the BabyAI-Visual domain in our paper. This environment uses visual observations, and our method successfully extends to this modality. We believe this provides strong evidence for the potential for IGE to be applied beyond text-based environments. Nonetheless, we recognize that further work is needed to validate its effectiveness in other types of environments, which we will make clear in the manuscript. We believe that will be extremely exciting future work!
> > >
> > > Thank you again!
> > >
> > > Best,
> > >
> > > The authors

---

### Author Response · Authors · 2024-11-22
**General Response (Part 1/2)**

We sincerely thank all the reviewers for their thoughtful and constructive feedback on our paper. We are delighted that our idea was found "original" and that it "creatively combines Go-Explore... and LM agent tasks" (Reviewer iSQ8), and that our method "shows strong performance over its baselines" (Reviewer iSQ8). We appreciate that our paper is "very clear and well written" (Reviewer xqm5), "fits into an important line of work" (Reviewer cN7x), and that our experiments "demonstrate significant performance gains" (Reviewer ZTB4).

**We have addressed all the concerns raised by the reviewers**, which has significantly improved the manuscript. Most importantly, we conducted new experiments demonstrating **(1)** that IGE's performance is robust to prompt engineering, addressing concerns about reliance on domain-specific prompts. **(2)** We also evaluated IGE with other foundation models, such as Claude Sonnet 3.5 and Llama-3 400B, showing that IGE consistently improves over baselines across different models. These new results confirm all previous findings and substantiate the claims regarding IGE's effectiveness and generality. Since all of the requested experiments only strengthen the case for IGE, we hope the reviewers will consider increasing their scores so IGE can be shared via a publication with the ICLR community.

More details on the major new experiments/additions:

1. **Robustness to Prompt Engineering**: We conducted experiments to show that IGE's success is independent of domain-specific prompt engineering. Specifically, we removed any hints on how to solve the tasks and kept the prompts purely factually descriptive about the environment. For example, as suggested by Reviewer ZTB4, in the BabyAI environment, we removed statements like "do not repeat actions if they have no effect" (full details in Appendix C.2). The results, shown in Table 1 below, indicate that even with minimal prompts, IGE **strongly outperforms baselines (statistically significant, p < 0.05)** with a domain-general prompt (save for a description of that environment). This demonstrates that IGE is flexible and general and can be applied easily out of the box to new domains, offering an improvement on that domain over the zero-shot capabilities of a foundation model. The evaluation setting is identical to that of Table 2 of our paper.

   **Table 1: IGE Performance with Domain-General Prompts**

   *Game of 24*

| Model                  | Naive LLM         | ReAct            | Reflexion        | IGE               |
|------------------------|-------------------|------------------|------------------|-------------------|
| Original (GPT-4)       | 0.70 ± 0.14       | 0.82 ± 0.12      | 0.83 ± 0.11      | 1.00 ± 0.00       |
| Domain-General Prompt (GPT-4) | 0.35 ± 0.13 | 0.71 ± 0.13      | 0.71 ± 0.13      | 0.96 ± 0.04       |

   *BabyAI "Put Next To"*

| Model                  | Naive LLM         | ReAct            | IGE               |
|------------------------|-------------------|------------------|-------------------|
| Original (GPT-4)       | 0.12 ± 0.12       | 0.48 ± 0.12      | 0.84 ± 0.08       |
| Domain-General Prompt (GPT-4) | 0.08 ± 0.12       | 0.24 ± 0.12      | 0.68 ± 0.12       |

   *TextWorld "Cooking Game"*

| Model                  | Naive LLM         | ReAct            | Reflexion        | IGE               |
|------------------------|-------------------|------------------|------------------|-------------------|
| Original (GPT-4)       | 0.40 ± 0.16       | 0.56 ± 0.16      | 0.52 ± 0.16      | 0.92 ± 0.08       |
| Domain-General Prompt (GPT-4) | 0.44 ± 0.16       | 0.48 ± 0.16      | 0.48 ± 0.16      | 0.72 ± 0.16       |

---

> ### Author Response · Authors · 2024-11-22
> **General Response (Part 2/2)**
>
> 2. **Performance with Different Foundation Models**: We evaluated IGE using other foundation models, including **Claude Sonnet 3.5** and **Llama-3 400B**, to address concerns about IGE's dependence on specific models. The results, presented in Table 2, show that IGE consistently outperforms baselines **(statistically significant, for all baselines with Llama3-400B and for all except Reflexion on Sonnet 3.5, p < 0.05)** across different models, demonstrating that IGE is a generic improvement that can be used in conjunction with any foundation model, likely including as new models are released that are even more powerful. The evaluation setting is identical to that of Table 2 of our paper. We have included these findings in Section 5 of the revised paper.
>
>    **Table 2: IGE Performance with Different Foundation Models on Game of 24**
>
> | Model                | Naive LLM         | ReAct            | Reflexion        | IGE               |
> |----------------------|-------------------|------------------|------------------|-------------------|
> | GPT-4                | 0.70 ± 0.12       | 0.82 ± 0.11      | 0.83 ± 0.10      | 1.00 ± 0.00       |
> | Claude Sonnet 3.5    | 0.24 ± 0.12       | 0.58 ± 0.14      | 0.80 ± 0.11      | 0.86 ± 0.09       |
> | Llama-3 400B         | 0.44 ± 0.14       | 0.68 ± 0.13      | 0.54 ± 0.14      | 0.98 ± 0.03       |
>
> 3. **Clarification of "Serendipitous Discoveries"**: We have provided a precise definition of what we mean by "serendipitous discoveries" in the context of IGE and explained how IGE enables discovering and leveraging unexpected but useful states during exploration, without prior knowledge from the human to craft predefined heuristics. **We have clarified this definition in the revised paper, specifically in Section 1 (Introduction), where we define serendipitous discoveries as 'states encountered during exploration that are valuable in terms of exploration, yet where what makes them interesting was not anticipated by the human user.'**
>
> 4. **Additional Clarifications and Discussions**: In response to specific concerns, we have added explanations and discussions to improve the clarity of our paper. This includes addressing questions about IGE's extension to stochastic environments, comparisons with other methods like MCTS and GFlowNets, the applicability to continuous state-action spaces, and the potential for future research directions. These discussions have been added to Section 6 (Conclusion and Limitations) and Appendix F of the revised paper.
>
> These changes address all the major questions raised by the reviewers. We believe the paper is now significantly stronger, and we deeply thank all the reviewers for their valuable feedback. We hope that the improvements we have made address your concerns fully. If there are any further questions or comments, we would be happy to discuss them. We kindly ask you to consider increasing your scores in light of these improvements.

---

### Meta-Review · Area_Chair_MYaX · 2024-12-22

**Metareview:**

The paper proposes a method that uses language models to guide exploration in reinforcement learning. This addresses an important question, uses timely tools, and advances understanding and capability. The paper is written and motivated clearly and includes sufficient empirical evaluation, including ablations. Some concerns were raised, and some remain, regarding overly strong or broad claims made in the paper, in particular regarding its applicability, as well as omission of limitations. The authors have sufficiently but not fully addressed these.

**Additional Comments On Reviewer Discussion:**

Reviewers have raised important concerns and authors did a thorough job of addressing them, prompting some reviewers to significantly raise their scores.

---

### Decision · Program_Chairs · 2025-01-22

Accept (Poster)